# CheXGenBench: A Unified Benchmark For Fidelity, Privacy and Utility of Synthetic Chest Radiographs

## Abstract

Structured benchmarks have advanced text-conditional image generation for real-world imagery, however, no such benchmark exists for synthetic radiograph generation. Despite being a highly active area of research, existing studies continue adopting inconsistent evaluation protocols and lack a unified assessment of the three most critical criteria: generative fidelity, privacy risk, and downstream utility. To address these limitations, we introduce CheXGenBench, the ***first*** unified evaluation framework for synthetic chest radiograph generation that simultaneously assesses fidelity, privacy risks, and clinical utility across frontier text-to-image (T2I) generative models. Our evaluation protocol, comprising over 20 quantitative metrics, covers 11 leading T2I architectures with *plug-and-play* integration for newer models. Through a rigorous and fair evaluation protocol, we establish a new SoTA in synthetic chest X-ray generation. Furthermore, our results uncover several critical limitations in the applicability of current generative models, which include **(1)** even SoTA models struggle with long-tailed medical distributions, **(2)** models pose high privacy risks regardless of fidelity quality, and **(3)** synthetic data offers limited utility for downstream multimodal tasks. Drawing from these results, we propose concrete research directions to advance the field. Finally, we curate and release SynthCheX-75K, a high-quality synthetic dataset comprising 75K radiographs generated by our top-performing model (Sana 0.6B). The fine-tuned models and the SynthCheX-75K dataset would be released after acceptance, while the *anonymised* code is available at this URL.

## 1 Introduction

Recent advances in multi-modal generative models, particularly Text-to-Image (T2I) systems (Ramesh et al., 2021; Saharia et al., 2022; Hurst et al., 2024; Xie et al., 2025), have demonstrated remarkable capabilities in producing high-fidelity synthetic images that closely adhere to natural-language prompts. Central to this progress is the development of comprehensive, well-designed benchmarks that evaluate various aspects of generative performance. These benchmarks drive innovation by establishing standardised evaluation protocols and creating an equitable foundation for model comparison. The natural imaging domain has benefited from numerous such benchmarks, each meticulously assessing specific aspects and identifying limitations that researchers subsequently address in developing next-generation models. For example, MS-COCO dataset (Lin et al., 2014) has been established as a seminal benchmark for evaluating general performance across multiple tasks, with particular emphasis on text-guided image generation. Building upon this foundation, more specialised benchmarks have emerged to assess specific attributes such as compositional understanding (Huang et al., 2023; 2025; Ghosh et al., 2023), enabling more nuanced analysis of model capabilities. Despite the advancements in the natural imaging domain, there remains a significant gap in medical image analysis, and specifically in terms of benchmarking specialised tasks such as text-to-image generation.

**Benchmarking Medical Imaging AI:** Medical image analysis has made significant strides by benefiting from rapid advancements in artificial intelligence; however, its progress faces substantial constraints due to what has be characterised as a benchmarking crisis (Mahmood, 2025). The ultimate goal of AI applications in medicine remains the development of intelligent systems capable of supporting and potentially transforming clinical decision-making processes. Progress toward this objective is

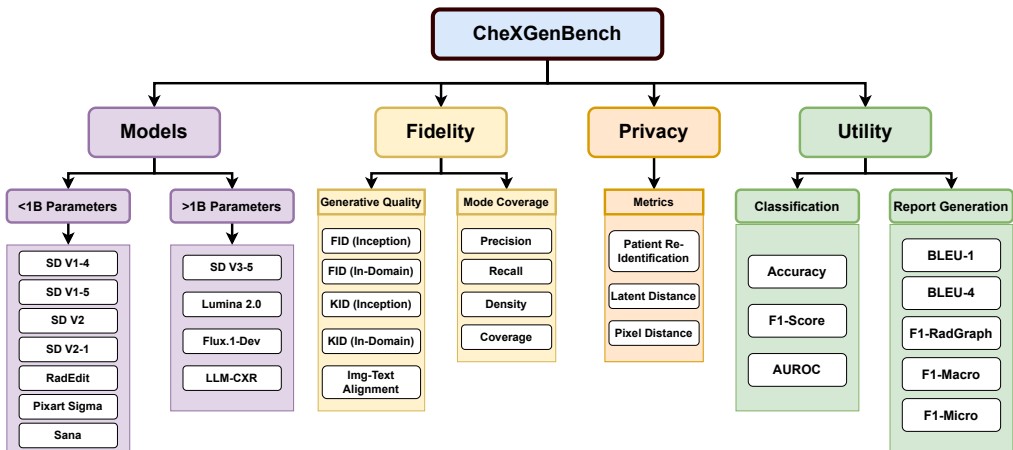

Figure 1: Figure illustrating the overall schematic of the CheXGenBench benchmark for evaluating text-to-image models in synthetic radiograph generation. CheXGenBench organises evaluation into three dimensions: **Fidelity** (measuring generative quality and mode coverage), **Privacy** (assessing memorisation and re-identification risks), and **Utility** (evaluating practical value of synthetic samples through image classification and radiology report generation metrics) through 20+ metrics across 11 widely-adopted **T2I models.**

frequently hindered by insufficient transparency in training and evaluation protocols, coupled with fragmented assessment criteria. Consequently, claims of "*state-of-the-art*" performance are often premature, non-reproducible, or reflect narrow contextual improvements rather than demonstrating genuine translational capability to clinical practice. The challenges of benchmarking in medical AI diagnostic tasks such as classification, localisation, segmentation and report generation have been discussed, and some improvements have been proposed (Irvin et al., 2019; Zong et al., 2023; Karargyris et al., 2023; Zhang et al., 2024). However, despite opportunities for impactful societal applications from diagnostic training to rare condition simulation, benchmarking in medical image *generation* remains in an even more nascent and unsatisfactory state. This is exacerbated by unique challenges such as unclear evalution metrics, data scarcity due to privacy constraints, and long-tailed distributions of rare pathologies.

While medical AI diagnostic tasks have been long studied, the generation of synthetic data through text prompts (medical descriptions) has recently emerged as a critical research focus. Besides being an interesting academic measure of medical AI capability, the ultimate motivation is that progress in diagnostic tasks is usually bottlenecked by *data scarcity* in the medical domain. High-fidelity generative models for synthetic data offer the promise to alleviate this bottleneck and ultimately facilitate clinical impact (Giuffrè & Shung, 2023). Chest radiographs are the most commonly used frontline modality in medical imaging. Although the majority of collected clinical data remain inaccessible behind institutional firewalls and compliances (Mahmood, 2025), several data repositories have been opened (Johnson et al., 2016; 2019; Irvin et al., 2019; Zhang et al., 2025), facilitating the development of generative models capable of synthesizing radiographs with potential relevance to downstream clinical utility (Ghalebikesabi et al., 2023; Bluethgen et al., 2024; Lee et al., 2024).

Current research on Text-to-Image generation of radiographs can be broadly categorized into two primary streams: **(1)** studies prioritizing the enhancement of image fidelity and generative performance (Lee et al., 2023; Weber et al., 2023; Lee et al., 2024; Dutt et al., 2024; Bluethgen et al., 2024; Han et al., 2024; Huang et al., 2024; Morís et al., 2024), and **(2)** investigations focusing on the mitigation of privacy concerns and patient re-identification risks (Fernandez et al., 2023; Dar et al., 2023; Akbar et al., 2025; Dutt et al., 2025; Dutt, 2025) that could undermine synthetic data utility. Despite constituting an active area of research with significant contributions, we have identified several critical benchmarking dimensions in which existing studies demonstrate consistent limitations. **Minimal or Absent Comparative Baselines:** Several notable studies either include minimal (self-proposed) (Dutt et al., 2024; 2025) or no baselines (Weber et al., 2023) for evaluation. (Han et al., 2024; Lee

et al., 2024) limit their comparative evaluations to only two competing methodologies. **Reporting Inadequate Metrics:** Studies throughout the existing literature predominantly report Fréchet Inception Distance (FID) (Heusel et al., 2017), specifically adopting in-domain image encoders based on primitive architectures (Cohen et al., 2022). Furthermore, no studies adequately report metrics characterising mode-coverage, a criterion of paramount importance in synthetic image generation for capturing the diversity of the underlying data distribution. Finally and most importantly, all metrics are reported as micro-averages across entire datasets without accounting for conditioning, let alone the long-tailed nature of medical data distributions. For example, excellent *average* generation fidelity could be reflective of the ability to generate more common images of healthy individuals, and the inability to generate images with pathologies of interest. This situation would offer limited clinical utility for pathology diagnosis.

**Restriction to Early-Stage Architectures:** Existing studies (Bluethgen et al., 2024; Weber et al., 2023; Dutt et al., 2024; Favero et al., 2025) have predominantly confined themselves to outdated T2I model architectures (Rombach et al., 2022), failing to address the crucial question of how recent advancements in generative modelling (Chen et al., 2024; Xie et al., 2025; Labs, 2024) from the natural imaging domain translate to the specialised requirements of medical imaging contexts. **Lack of a Unified Evaluation:** Existing studies are fragmented between evaluating generative fidelity (Bluethgen et al., 2024; Weber et al., 2023; Lee et al., 2024) or privacy and re-identification risks (Fernandez et al., 2023; Akbar et al., 2025; Dutt, 2025), failing to conduct and provide a unified evaluation of the two key aspects of synthetic radiograph generation research. **Limited Evaluation on Synthetic Data Utility:** Most studies fail to comment on the downstream utility of synthetic radiographs (Weber et al., 2023; Lee et al., 2024) often presenting generation results without rigorously assessing their potential impact on medical image analysis tasks such as classification, segmentation, or diagnostic reasoning. This need for rigour is reflected in recent standardisation efforts, such as the "Scorecards" for synthetic medical data proposed by (Zamzmi et al., 2024), which reports on key dimensions like fidelity, utility, and privacy.

To address these critical limitations, we introduce CheXGenBench, a comprehensive benchmark designed for rigorous evaluation of frontier generative models across a diverse spectrum of metrics encompassing: (1) generation fidelity and mode coverage, (2) privacy and re-identification risk assessment, and (3) downstream clinical utility through an extensive suite of 20+ quantitative and interpretable metrics. We establish standardised training and evaluation protocols to enable equitable comparison between diverse model architectures. Furthermore, our work is highly complementary to standardised evaluation approaches such as model and data card (Zamzmi et al., 2024) by providing the information required to populate those cards. CheXGenBench prioritises usability and adaptability, facilitating seamless plug-and-play integration of both existing and emerging generative frameworks. Through systematic evaluation, we present several T2I models previously **unexplored** for radiograph generation and establish new state-of-the-art (SoTA) performance. Furthermore, we release a synthetic dataset, *SynthCheX-75K* , comprising 75K high-quality radiographs generated by our benchmark-leading model to facilitate advancement in medical image analysis research.

## 2 CHEXGENBENCH DESIGN

CheXGenBench is designed to evaluate each text-to-image model across three dimensions comprehensively: **(1)** generative fidelity and mode coverage (Sec 2.2), **(2)** privacy and patient re-identification risks (Sec 2.2), and **(3)** synthetic data utility for downstream tasks (Sec 2.4), which we elucidate in detail in the following sub-sections.

**Design Principles:** To maximise usability, CheXGenBench features *decoupled* training and evaluation pipelines. This allows researchers to use their preferred training frameworks (e.g., Hugging Face Diffusers (von Platen et al., 2022)) and then automatically assess their models on over 20 standardized metrics by simply providing the generated images and a metadata file.

### 2.1 TRAINING DATASET AND PROTOCOL

**Criteria for Model Inclusion:** Our model selection was guided by two main criteria. First, we included models previously used for synthetic radiograph generation in existing literature (Rombach

et al., 2022; Bluethgen et al., 2024; Lee et al., 2024; Pérez-García et al., 2024) to provide continuity and comparability with prior work. Second, we incorporated recent state-of-the-art models (both diffusion and auto-regressive) (Chen et al., 2024; Esser et al., 2024; Xie et al., 2025; Qin et al., 2025; Labs, 2024) from the natural imaging domain that had not yet been evaluated for chest X-ray synthesis. Thus we both benchmark established methods and assess the potential of newer architectures for medical image generation.

The models in our benchmark were stratified into two categories based on parameter count: (1) models with fewer than 1B parameters, and (2) models exceeding 1B parameters. For the smaller models, we employed full fine-tuning (FFT) of all parameters. For larger architectures, we implemented Low-Rank Adaptation (LoRA) (Hu et al., 2022) with rank 32 on standard query, key, and value layers in the attention blocks to address both computational constraints and reflect realistic training scenarios. In Appendix I, we present an ablation study that investigates the impact of increasing the LoRA rank for these models.

**Training Dataset with Enhanced Captions:** All evaluations are conducted on the MIMIC-CXR dataset (Johnson et al., 2019), which has become the de-facto standard for text-to-image radiograph generation (Bluethgen et al., 2024; Weber et al., 2023; Pérez-García et al., 2024; Lee et al., 2024; Dutt et al., 2024). While prior work has often relied on abbreviated captions derived from rule-based methods (Pérez-García et al., 2024; Bluethgen et al., 2024), these approaches can be inadequate due to inconsistencies in clinical terminology and report structure (Zambrano Chaves et al., 2025). Reflecting a recent shift towards deep learning techniques for generating more comprehensive summaries (Segalis et al., 2023; Zambrano Chaves et al., 2025), and motivated by evidence that more descriptive captions enhance generative fidelity, we are the first study to adopt the enhanced "LLaVA-Rad" annotations for this task. We empirically validate this choice in Appendix B, where we demonstrate that using these annotations leads to improved image fidelity and reduced re-identification risks, and thus we strongly recommend them for future research in this domain.

**Training Protocol:** Our analysis revealed that prior studies employed inconsistent training budgets, undermining valid cross-model comparison. To establish a level evaluation framework, we implemented a standardised training protocol across all T2I models. Each architecture was trained for precisely 20 epochs on an identical training split of 237,388 samples annotated with "LLaVA-Rad" annotations. We release the training and evaluation data splits along with the benchmark.

## 2.2 EVALUATING GENERATIVE FIDELITY, MODE COVERAGE, AND CONDITIONAL ANALYSIS

**Limitations of Current Fidelity Assessment:** The Fréchet Inception Distance (FID) score (Heusel et al., 2017) is widely used for evaluating synthetic radiograph fidelity. Standard FID uses InceptionV3 (Szegedy et al., 2016) features trained on natural images, creating domain mismatch for medical images (Kynkäänniemi et al., 2023). While some research employs domain-specific encoders like DenseNet-121 (Huang et al., 2017) trained on radiographs (Cohen et al., 2022), we argue these adaptations remain inadequate. DenseNet-121, despite domain alignment, represents an outdated backbone that fails to capture critical nuances, reducing FID reliability for radiograph quality assessment. We perform an extensive ablation on this in Appendix Section C. Additionally, studies using natural image-text pre-trained CLIP (Radford et al., 2021; Bluethgen et al., 2024) further compromise evaluation integrity due to high domain misalignment.

**Improving metric Reliability in CheXGenBench :** We address the aforementioned limitations and enable a more nuanced evaluation of synthetic radiographs in CheXGenBench . For robust image fidelity assessment (FID, KID, Precision, Density, Recall, Coverage), CheXGenBench employs features from *RadDino* (Pérez-García et al., 2025), a SoTA model trained on 838K X-rays, which acts as a highly-capable feature extractor for X-rays and significantly improves metric reliability over prior works. We also report image-text alignment ("Alignment Score"), a reference-free metric functioning as a domain-specific CLIPScore (Hessel et al., 2021). We utilize *BioViL-T* (Bannur et al., 2023) to compute the global cosine similarity between the generated image and text prompt embeddings within a shared semantic space, quantifying their semantic consistency .

**Expanding the Metric Suite with Density and Mode Coverage:** All prior studies have reported generation fidelity without a systematic evaluation of how effectively generated samples capture critical distributional characteristics, notably the density of the resulting sample distribution and coverage of distinct modes from the true data. This is of particular importance in medical datasets

since not all pathologies are distributed equally. To address this, CheXGenBench also supports Precision, Recall, Density and Coverage (Naeem et al., 2020).

Precision, Recall, Density, and Coverage (PRDC) metrics are essential for evaluating mode coverage, particularly in long-tailed distributions where FID scores can be skewed by majority classes. This is evident in the MIMIC-CXR dataset, where "No Finding" radiographs, representing healthy X-rays, predominate despite pathological images offering greater clinical value for synthesis. Within the PRDC framework, **Precision** quantifies generated sample realism, **Recall** measures how well the real distribution is captured, **Density** assesses feature space concentration, and **Coverage** determines the proportion of modes of real data successfully generated, together providing a more comprehensive assessment than global metrics alone.

**Conditional Analysis for Individual Pathologies:** CheXGenBench extends evaluation capabilities through pathology-specific conditional analysis, wherein we compute each generation fidelity metric independently across individual pathologies in the MIMIC-CXR dataset. This granular assessment approach provides critical insights that enable developers to precisely evaluate generative performance for specific medical conditions. Such an analysis becomes particularly valuable in scenarios requiring selective augmentation of underrepresented pathologies through a generative model, which is the most desired use case. Our framework calculates FID, KID, image-text alignment, and PRDC metrics for each distinct pathology, facilitating comprehensive performance evaluation at the condition-specific level. To the best of our knowledge, we are the first study to include pathology-conditional evaluation.

## 2.3 EVALUATION PROTOCOL: PRIVACY AND PATIENT RE-IDENTIFICATION

Deep generative models can inadvertently memorise distinctive training examples, allowing an attacker to reverse–engineer sensitive patient information from seemingly "synthetic" images (Carlini et al., 2023; Jegorova et al., 2023). In the medical domain, even coarse anatomical cues may be enough to link a generated radiograph back to an individual, breaching data-protection regulations such as HIPAA[1] and the EU GDPR[2]. Consequently, to assess clinical relevance claims, a benchmark that **must** characterise (i) how much a model memorises, and (ii) how easily a real patient can be re-identified from its outputs.

**Re-identification risk formulation.** To evaluate privacy and patient re-identification risks, we implement established metrics from the existing literature (Fernandez et al., 2023; Akbar et al., 2025; Dutt et al., 2025; Dutt, 2025). Let $\mathcal{D}_{\text{real}} = \{x_i, c_i\}_{i=1}^N$ be the training set of chest radiographs $x_i$ with corresponding captions $c_i$, and let $G_\theta$ denote a text-to-image model producing synthetic images $\hat{x} = G_\theta(c)$. We assess whether a generated sample $\hat{x}$ memorizes any training image $x_i$ via a learned similarity function $\ell(\hat{x}, x_i)$ with $\text{Memorised}(x_i) \Leftrightarrow \ell(\hat{x}, x_i) > \delta$, where $\delta$ is a fixed safety margin. We evaluate similarity through three distinct lenses: (i) **Pixel distance** ($\ell_{\text{pix}}$): $\|\hat{x} - x_i\|_2$, to detect near-duplicates (Carlini et al., 2023). (ii) **Latent distance** ($\ell_{\text{lat}}$): Normalized Euclidean distance in the embedding space of *RadDino* (Pérez-García et al., 2025). (iii) **Re-identification score** ($s_{\text{reid}}$): Probability that $\hat{x}$ and $x_i$ are from the same patient, as estimated by a Siamese neural network (Packhäuser et al., 2022) The privacy metrics are further formalised in Appendix M.

**Assessing re-identification.** Instead of relying solely on pixel-based (Carlini et al., 2023) or structural-based (Kumar et al., 2017) similarity, we adopt the deep learning-based metric $\ell = f_\theta^{\text{re-id}}$ (Packhäuser et al., 2022). The model $f_\theta^{\text{re-id}}$ is a Siamese network with a ResNet-50 (He et al., 2016) backbone trained to classify whether two chest X-ray images originate from the same patient. For any pair $(\hat{x}, x)$ of a generated and real image, $f_\theta^{\text{re-id}}$ outputs a re-identification score $s_{\text{reid}} \in [0, 1]$ after a sigmoid layer. A synthetic image $\hat{x}$ is considered re-identified if $s_{\text{reid}} \geq \delta$ for any training image $x$. Acknowledging that any single DL-based metric can be unreliable, CheXGenBench provides a more robust privacy assessment by using pixel distance ($\ell_{\text{pix}}$) and latent distance ($\ell_{\text{lat}}$) as complementary evaluation methods.

## 2.4 EVALUATION PROTOCOL: SYNTHETIC DATA UTILITY FOR DOWNSTREAM TASKS

A significant application of synthetic data in radiology lies in enhancing downstream model performance, potentially circumventing the stringent sharing restrictions imposed on medical datasets. For

---

[1]https://www.hhs.gov/hipaa/for-professionals/privacy/index.htm

[2]http://gdpr.eu/what-is-gdpr/

Table 1: Table comparing the results for generative fidelity for 11 different T2I models in the benchmark. The best result for each metric is indicated with **bold**, while the second-best result is underlined. The overall best performing model (See Appendix Tab. 8) is highlighted in green.

| Model | Size | Default Resolution | Prev. Available For CXR? | Fine-Tuning | FID ↓ (RadDino) | KID ↓ (RadDino) | Alignment Score ↑ | Precision ↑ | Recall ↑ | Density ↑ | Coverage ↑ |
|---|---|---|---|---|---|---|---|---|---|---|---|
| **SD V1-4** (Rombach et al., 2022) | 0.86B | 512 | ✓ | FFT | 125.186 | 0.172 | 0.357 | 0.488 | 0.301 | 0.236 | 0.217 |
| **SD V1-5** (Rombach et al., 2022) | 0.86B | 512 | ✓ | FFT | 118.932 | 0.147 | 0.326 | 0.536 | 0.473 | 0.242 | 0.256 |
| **SD V2** (Rombach et al., 2022) | 0.86B | 512 | ✓ | FFT | 194.724 | 0.376 | 0.311 | 0.480 | 0.086 | 0.166 | 0.057 |
| **SD V2-1** (Rombach et al., 2022) | 0.86B | 512 | ✓ | FFT | 186.530 | 0.413 | 0.197 | 0.530 | 0.049 | 0.180 | 0.038 |
| **RadEdit** (Pérez-García et al., 2024) | 0.86B | 512 | ✓ | N/A | 69.695 | 0.033 | 0.677 | 0.397 | 0.544 | 0.150 | 0.285 |
| **Pixart Sigma** (Chen et al., 2024) | 0.60B | 512 | ✗ | FFT | 60.154 | 0.023 | **0.697** | 0.666 | 0.522 | 0.506 | 0.506 |
| **Sana** (Xie et al., 2025) | 0.60B | 512 | ✗ | FFT | **54.225** | **0.016** | 0.695 | 0.674 | **0.614** | 0.520 | **0.548** |
| **SD V3.5 Medium** (Esser et al., 2024) | 2.50B | 1024 | ✗ | LoRA(r=32) | 91.302 | 0.103 | 0.044 | 0.632 | 0.205 | 0.401 | 0.244 |
| **Lumina 2.0** (Qin et al., 2025) | 2.60B | 1024 | ✗ | LoRA(r=32) | 101.198 | 0.110 | 0.121 | 0.574 | 0.014 | 0.256 | 0.170 |
| **Flux.1-Dev** (Labs, 2024) | 12B | 1024 | ✗ | LoRA(r=32) | 122.400 | 0.144 | 0.036 | 0.420 | 0.008 | 0.125 | 0.326 |
| **LLM-CXR** (Lee et al., 2024) | 12B | 256 | ✓ | N/A | 71.243 | 0.061 | 0.319 | **0.782** | 0.041 | **0.671** | 0.459 |

our utility assessment framework, we have strategically selected two prevalent downstream tasks previously established in radiological evaluation (Bluethgen et al., 2024): **(1) Image Classification** and **(2) Radiology Report Generation (RRG)**. These tasks were deliberately chosen for their distinct complexities. Image classification serves as a unimodal assessment, directly evaluating the intrinsic quality of synthetic radiographs, while RRG functions as a more demanding multimodal evaluation, assessing the factual correctness between synthetic images and their corresponding clinical descriptions. **Note that** other downstream tasks, such as segmentation and localisation were deliberately excluded due to the absence of relevant ground-truth annotations in the MIMIC-CXR dataset.

**Downstream Image Classification:** We adopt the experimental setting previously utilised (Bluethgen et al., 2024) and measure classification performance when a classifier (He et al., 2016) is trained *exclusively* on synthetic data (20K samples) ($\mathcal{D}_{syn}$) (for 20 epochs). This provides us with an idea of the stand-alone clinical value of the synthetic data from a generative model. The performance metrics are calculated on a held-out real test set ($\mathcal{D}_{test}$) to ensure clinical relevance and generalizability of our findings. We quantify classification performance through standard accuracy, F1-Score, and AUROC metrics.

**Downstream Report Generation:** We choose an existing multimodal model *LLaVA-Rad* (Zambrano Chaves et al., 2025) with SoTA radiograph understanding abilities and continue to fine-tune it with 20,000 synthetic samples. The performance is reported on a real test set ($\mathcal{D}_{test}$). We quantify RRG performance using the standard metrics adopted in literature (Bluethgen et al., 2024; Zambrano Chaves et al., 2025): BLEU (Papineni et al., 2002), ROUGE-L (Lin, 2004), F1-RadGraph (Jain et al., 2021), and F1-Score. Additionally, we also incorporate newer metrics GREENScore (Ostmeier et al., 2024) and RaTEScore (Zhao et al., 2024). BLEU and ROUGE-L assess lexical overlap and fluency, while F1-RadGraph and F1-Scores (5/14) evaluate factual correctness and specific findings. GREENScore and RaTEScore leverage language models to capture semantic nuances and sensitivity to negations.

## 3 EXPERIMENTS AND RESULTS

**Training Setting:** Our evaluation used various T2I models with different training approaches. We used existing radiograph generation models (RadEdit (Pérez-García et al., 2024), LLM-CXR (Lee et al., 2024)) unmodified. For models under 1B parameters, we performed full fine-tuning (FFT), while larger models used Parameter-Efficient Fine-Tuning (PEFT) with LoRA (Hu et al., 2022) (rank 32) on query, key, and value layers. Stable-Diffusion variants utilised Huggingface Diffusers (von Platen et al., 2022), while Sana and Pixart-Sigma were trained using their official repositories. Larger models employed the ai-toolkit package[3]. Training followed officially recommended hyperparameters with a consistent total batch size of 128 across 4 Nvidia H200 GPUs, using 237,388 training samples and 5,034 test samples. Downstream evaluation experiments were conducted on Nvidia A100 GPUs.

### 3.1 FIDELITY OF SYNTHETIC GENERATIONS

**Results on Global Assessment**

---

[3]https://github.com/ostris/ai-toolkit

Table 2: Comparison of FID (RadDino) scores (↓) across individual pathologies in the MIMIC-CXR dataset. Lower values indicate superior performance, with the best results for each pathology highlighted in **bold**. The most challenging pathology (highest average FID across models) is highlighted in red , while the best-performing model overall is highlighted in green . We also highlight the best performing pathology for each model in blue .

| Model | Atelectasis | Cardiomegaly | Consolid. | Edema | EC | Fracture | LL | LO | NF | PE | PO | PN | PT | SD |
|---|---|---|---|---|---|---|---|---|---|---|---|---|---|---|
| SD V1-4 | 134.11 | 131.04 | 184.30 | 144.84 | 217.75 | 238.78 | 225.99 | 129.38 | 106.34 | 128.16 | 255.84 | 163.82 | 212.48 | 135.10 |
| SD V1-5 | 125.67 | 124.75 | 181.25 | 139.94 | 213.94 | 243.17 | **123.13** | 255.64 | 167.81 | 193.75 | 243.17 | 101.08 | **119.91** | 123.64 |
| SD V2 | 188.72 | 193.91 | 241.24 | 214.40 | 214.40 | 253.91 | 268.28 | 280.11 | 193.99 | 299.48 | 223.43 | 250.96 | 183.34 | 193.99 |
| SD V2-1 | 179.20 | 181.79 | 228.43 | 193.62 | 242.65 | 263.01 | 260.15 | 185.00 | 192.30 | 178.84 | 287.27 | 213.26 | 242.60 | 176.99 |
| RadEdit | 63.38 | 62.79 | 136.59 | 76.94 | 155.97 | 197.58 | 184.11 | 61.90 | 67.88 | 60.60 | 215.92 | 114.66 | 151.34 | 53.10 |
| Pixart Sigma | 59.27 | 60.39 | 133.96 | 73.93 | 155.53 | 179.44 | 174.63 | 56.83 | 48.74 | 59.05 | 210.90 | 108.42 | 150.55 | 51.61 |
| Sana | **51.03** | **54.68** | **127.46** | **67.84** | **147.00** | **172.32** | 163.14 | **49.23** | **44.60** | **49.80** | **199.45** | **99.52** | 141.99 | **46.51** |
| SD V3.5 Medium | 94.94 | 94.84 | 149.05 | 111.94 | 168.48 | 184.75 | 173.37 | 86.72 | 89.60 | 91.92 | 203.62 | 124.07 | 163.27 | 86.99 |
| Lumina 2.0 | 109.39 | 111.11 | 162.36 | 131.18 | 182.35 | 191.83 | 182.22 | 99.53 | 95.66 | 105.25 | 213.50 | 134.58 | 165.09 | 102.78 |
| Flux.1-Dev | 137.10 | 133.60 | 176.76 | 152.91 | 191.48 | 191.02 | 194.97 | 133.37 | 100.58 | 137.66 | 221.23 | 156.59 | 190.93 | 127.03 |
| LLM-CXR | 71.57 | 71.37 | 136.65 | 83.18 | 148.28 | 168.50 | 163.22 | 66.93 | 64.62 | 67.83 | 200.84 | 108.04 | 147.52 | 67.54 |

The results are presented in Tab. 1, where we showcase both fidelity and mode coverage metrics. Sana (Xie et al., 2025) delivers superior overall performance across key metrics, achieving the lowest FID and KID scores, indicating exceptional generation fidelity, while simultaneously attaining the highest Recall and Coverage, demonstrating its capacity to capture diverse modes (distributions) throughout the dataset. Pixart Sigma (Chen et al., 2024) emerges as a strong contender, exhibiting the highest image-text alignment alongside second-best FID, KID, and Coverage scores. LLM-CXR (Lee et al., 2024) exhibits specialised capabilities, substantially outperforming all other models in Precision; however, its notably low Recall suggests limited generative scope, primarily effective for specific distributions (pathologies). This also highlights that conventional fidelity metrics like FID do not present a complete picture of the model performance. Earlier Stable-Diffusion variants (SD V1-x, V2-x) demonstrate consistently suboptimal scores across all metrics despite full fine-tuning, a particularly significant finding given their prevalent adoption in the synthetic radiograph generation literature (Bluethgen et al., 2024; Favero et al., 2025; Fernandez et al., 2023; Dutt et al., 2024). Larger architectural models (SD V3-5 (Esser et al., 2024), Lumina 2.0 (Qin et al., 2025), Flux.1-Dev (Labs, 2024)), with the exception of LLM-CXR, yield predominantly inferior performance across evaluation metrics. SD V3-5 exhibits high Precision but low Recall, Density, and Coverage scores, mirroring trends observed in LLM-CXR. We hypothesise that this stems from the inability of LoRA to provide sufficient adaptation for the medical domain, a limitation previously observed in (Dutt et al., 2024). Performance improvements might be achievable by extending LoRA to other linear layers beyond attention (Q,K,V) layers, however, we leave this exploration to future work. Overall, Sana achieves the optimal performance-efficiency trade-off among all evaluated models. **Notably**, Sana has been adapted for synthetic radiograph generation for the first time through this work.

**Results on Conditional Assessment**

Results for conditional analysis on individual pathologies are presented in Tab. 2. Consistent with trends observed in Tab. 1, Sana demonstrates superior performance, achieving the lowest FID scores across 12 of the 14 pathology categories. This indicates Sana's robust capability to generate high-fidelity radiographs across diverse pathological conditions. Pixart Sigma maintains its position as the second-best performing model, while RadEdit frequently secures the third-best scores across multiple categories. LLM-CXR demonstrates competitive performance for specific pathologies, notably achieving strong results for Edema (83.18) and No Finding (64.62), frequently outperforming both earlier Stable Diffusion models and certain large-scale models (SD V3.5, Lumina, Flux.1-Dev).

**Concerning Observations:** This analysis reveals concerning patterns. Substantial performance variation exists across pathologies for each model, regardless of overall performance. For instance, Sana exhibits FID scores ranging from 44.60 for "*No Finding (NF)*" to 199.45 for "*Pleural Other (PO)*". Notably, five of the eleven models achieve optimal performance on "No Finding" cases, which represent healthy radiographs with limited clinical utility from synthetic X-rays, while all models consistently perform poorly on "*Pleural Other (PO)*" pathology. In Appendix E and Tab. 10, we empirically demonstrate that model performance strongly correlates with pathology prevalence in the

---

[1]*Note:* **EC** = Enlarged Cardiomediastinum, **LL** = Lung Lesion, **LO** = Lung Opacity, **NF** = No Finding, **PE** = Pleural Effusion, **PO** = Pleural Other, **PN** = Pneumonia, **PT** = Pneumothorax, **SD** = Support Devices.

Table 3: Results on Re-Identification Risk and Patient Privacy Metrics. We present the average scores across 2000 samples and individual scores with maximum privacy risks.

| Model | SD V1-4 | SD V1-5 | SD V2 | SD V2-1 | RadEdit | Sana | Pixart-$\Sigma$ | SD V3-5 | Lum. 2.0 | Flux | LLM-CXR |
|---|---|---|---|---|---|---|---|---|---|---|---|
| Avg. Re-ID Score ($\downarrow$) | 0.539 | 0.572 | 0.533 | 0.503 | 0.481 | 0.551 | 0.548 | **0.365** | 0.513 | 0.404 | 0.537 |
| Avg. Latent Distance ($\uparrow$) | 0.592 | 0.583 | 0.588 | 0.592 | 0.560 | 0.540 | 0.561 | **0.601** | 0.591 | 0.595 | 0.557 |
| Avg. Pixel Distance ($\uparrow$) | 143 | 143 | 143 | 145 | 145 | **162** | 159 | 147 | 145 | 155 | 149 |
| Max. Re-ID Score ($\downarrow$) | 0.996 | 0.996 | 0.996 | 0.997 | 0.992 | 0.996 | 0.994 | 0.997 | 0.993 | 0.992 | 0.994 |
| Count Re-ID > $\delta$ ($\downarrow$) | 434 | 498 | 454 | 392 | 380 | 442 | 442 | 236 | 223 | 196 | 419 |

training dataset (**correlation coefficient: 0.947**), suggesting that current models mainly reproduce the largest modes in the dataset, while failing to model the longer tail of pathologies, and thus fail to achieve general clinical utility. Thus future medical image generation models should follow this evaluation protocol in order to make claims of clinical utility. We also hope this analysis encourages developers to incorporate training strategies tailored for long-tailed distributions (Qin et al., 2023).

## 3.2 RESULTS ON PRIVACY AND RE-IDENTIFICATION RISKS

**Experimental Setting:** To quantify re-identification risks, we select a subset of 2000 image-text pairs $(x_i^{img}, x_i^{txt})$ from the training set and generate $N(=10)$ synthetic samples $\hat{x}_i^{img,1}, \hat{x}_i^{img,2}, \ldots, \hat{x}_i^{img,N}$ using 10 different random seeds for each training prompt. Next, we calculate Re-ID scores, Pixel and Latent Distances between each real-synthetic pair $(x_i^{img}, \hat{x}_i^{img,n})$ for all $n \in \{1, 2, \ldots, N\}$. Finally, we report the maximum Re-ID score $\max_j s_{\text{reid}}^{(j)}$, minimum pixel distance $\min_j \ell_{\text{pix}}^{(j)}$ and minimum latent distance $\min_j \ell_{\text{lat}}^{(j)}$ across $N$ generations for each sample. This approach enables us to identify the greatest privacy risk posed for each training sample across multiple generations.

**Results:** The privacy risk assessment results are detailed in Tab. 3. Most models exhibit Average Re-ID scores within a comparable range, with SD V3-5 notably achieving the lowest (most favourable) score. For latent and pixel distances, a similar pattern emerges, where SD V3-5 and Sana demonstrate superior performance (i.e., lower average distances), respectively. **Concerns:** We conducted a detailed analysis of *individual* scores across 2,000 samples, with particular attention to two key metrics: (1) the maximum Re-ID score, which represents the highest potential for re-identification, and (2) the frequency of samples exceeding a high-risk threshold ($\delta = 0.85$). Our results reveal that all models, regardless of their fidelity performance, generate samples that can be re-identified with high confidence. The proportion of samples presenting significant re-identification risk remains substantial across all models, ranging from 10% to 25%. Notably, models trained with LoRA demonstrate a relatively lower incidence of high-risk samples, potentially due to their reduced capacity for memorization (Dutt et al., 2025). These findings underscore a critical insight: **generative models pose substantial privacy risks irrespective of their generative capabilities**.

## 3.3 UTILITY OF SYNTHETIC SAMPLES FOR DOWNSTREAM TASKS

**Downstream Image Classification:** We present the results in Tab. 4. Sana emerges as exceptionally effective, with its synthetic images enabling classifiers to match or exceed the original data baseline on an impressive 10 out of 13 pathologies. Interestingly, it surpasses the baseline on *Fracture*, an under-represented class in MIMIC-CXR. Other models, such as RadEdit, Pixart-Sigma, and LLM-CXR, show limited success by matching the baseline for at most two pathologies, failing to surpass it. Models like SD V1-4, SD V3-5, Lumina 2.0, and Flux.1-Dev generally produce synthetic data that leads to classifiers significantly underperforming the Original Data baseline across most or all pathologies. **Viability:** The results from Sana strongly suggest that high-quality synthetic data can, in some cases, be a viable standalone replacement for real data for training medical image classifiers. This is a powerful finding with implications for data privacy, scarcity, and augmentation. In Appendix D, we expand on the correlation between generative fidelity and downstream utility for classification.

**Downstream Radiology Report Generation** The results are presented in Tab. 5. Firstly, we observe that *additional* fine-tuning with synthetic data, irrespective of the model, leads to a performance degradation as compared to the original baseline (trained with real data). In terms of the models,

Table 4: Performance Comparison (AUC ↑) of a ResNet50 classifier trained **only on synthetic data** vs. Original (Real) Data Baseline for all pathologies in the MIMIC dataset. Results matching or exceeding the Original Data baseline are **bolded** and within 0.01 AUC are underlined.

| Model | Atel-ectasis | Cardio-megaly | Consol-idation | Edema | EC | Fracture | LL | LO | NF | PE | PO | PN | PT | SD |
|---|---|---|---|---|---|---|---|---|---|---|---|---|---|---|
| **Original (Real)** | 0.75 | 0.76 | 0.72 | 0.85 | 0.61 | 0.58 | 0.63 | 0.70 | 0.84 | 0.84 | 0.74 | 0.67 | 0.71 | 0.83 |
| SD V1-4 | 0.70 | 0.70 | 0.67 | 0.81 | 0.56 | 0.57 | 0.63 | 0.67 | 0.80 | 0.77 | 0.65 | 0.60 | 0.65 | 0.80 |
| SD V1-5 | 0.72 | 0.72 | 0.69 | 0.81 | 0.60 | 0.53 | **0.66** | 0.67 | 0.82 | 0.79 | 0.68 | 0.62 | 0.70 | **0.83** |
| SD V2 | 0.66 | 0.69 | 0.66 | 0.78 | **0.61** | 0.53 | 0.55 | 0.63 | 0.75 | 0.76 | 0.50 | 0.61 | 0.64 | 0.78 |
| SD V2-1 | 0.63 | 0.67 | 0.65 | 0.71 | 0.55 | **0.59** | 0.62 | 0.62 | 0.75 | 0.74 | 0.57 | 0.56 | 0.61 | 0.75 |
| RadEdit | 0.73 | 0.73 | **0.72** | 0.84 | **0.61** | 0.56 | 0.60 | 0.69 | 0.81 | 0.82 | 0.72 | 0.66 | 0.66 | 0.77 |
| Pixart Sigma | 0.74 | 0.73 | 0.70 | 0.84 | **0.61** | **0.58** | 0.61 | 0.69 | 0.83 | 0.81 | 0.68 | 0.63 | 0.70 | 0.80 |
| Sana | 0.74 | **0.76** | **0.72** | **0.85** | **0.61** | **0.62** | **0.63** | **0.70** | 0.83 | **0.84** | 0.73 | 0.64 | **0.72** | **0.83** |
| SD V3-5 | 0.55 | 0.55 | 0.56 | 0.55 | 0.47 | 0.47 | 0.47 | 0.53 | 0.60 | 0.54 | 0.58 | 0.49 | 0.55 | 0.71 |
| Lumina 2.0 | 0.46 | 0.48 | 0.52 | 0.51 | 0.46 | 0.57 | 0.53 | 0.52 | 0.59 | 0.55 | 0.57 | 0.49 | 0.50 | 0.71 |
| Flux.1-Dev | 0.41 | 0.41 | 0.44 | 0.40 | 0.44 | 0.52 | 0.48 | 0.42 | 0.40 | 0.38 | 0.50 | 0.48 | 0.44 | 0.67 |
| LLM-CXR | 0.70 | 0.69 | 0.70 | 0.81 | **0.61** | 0.57 | 0.54 | 0.65 | 0.80 | 0.77 | 0.66 | 0.61 | 0.63 | 0.73 |

Table 5: Radiology Report Generation (RRG) performance metrics for various generative models.

| Metric | Original | SD V1-4 | SD V1-5 | SD V2 | SD V2-1 | RadEdit | Pixart-Σ | Sana | SD V3-5 | Lumina 2.0 | Flux.1-Dev | LLM-CXR |
|---|---|---|---|---|---|---|---|---|---|---|---|---|
| **BLEU-1** (↑) | **38.16** | 25.85 | 26.02 | 26.62 | 26.76 | 30.55 | **31.25** | 31.11 | 23.49 | 17.96 | 18.19 | 29.78 |
| **BLEU-4** (↑) | **15.38** | 6.76 | 7.50 | 7.35 | 7.38 | **8.36** | 7.91 | 7.70 | 4.91 | 4.27 | 3.36 | 7.93 |
| **ROUGE-L** (↑) | **0.31** | 0.23 | **0.24** | **0.24** | **0.24** | **0.24** | 0.23 | **0.24** | 0.20 | 0.20 | 0.18 | 0.23 |
| **F1-RadGraph** (↑) | **0.29** | 0.21 | **0.24** | 0.23 | 0.22 | **0.24** | 0.22 | 0.23 | 0.17 | 0.18 | 0.14 | 0.21 |
| **Micro F1-5** (↑) | **0.57** | 0.56 | 0.54 | **0.57** | **0.57** | 0.55 | 0.50 | **0.57** | 0.31 | 0.41 | 0.32 | 0.55 |
| **Micro F1-14** (↑) | **0.57** | 0.53 | 0.51 | **0.55** | 0.53 | **0.55** | 0.52 | **0.55** | 0.35 | 0.41 | 0.36 | 0.53 |
| **GREENScore** (↑) | **0.36** | 0.28 | 0.27 | 0.27 | 0.27 | 0.30 | 0.30 | **0.31** | 0.28 | 0.28 | 0.27 | 0.30 |
| **RaTEScore** (↑) | **0.58** | 0.46 | 0.46 | 0.44 | 0.43 | 0.49 | 0.50 | **0.51** | 0.45 | 0.45 | 0.44 | 0.50 |

RadEdit and Sana emerge as leading performers. RadEdit excels in BLEU-4 (fluency) and is a top contender in F1-RadGraph and Micro F1-14, while Sana demonstrates strengths in ROUGE-L, Micro F1, GREENScore and RaTEScore. LLM-CXR also gives a strong performance, often giving second- or third-best scores. Pixart Sigma shows the best individual word usage (BLEU-1) and GREENScore, and models like SD V2 also perform well in Micro F1 scores. Overall, no single model dominates in all metrics but Sana consistently emergest as the top-performer in several metrics. **Concerns:** These results reflect that current T2I models might show high fidelity, but their utility for multimodal tasks such as report generation is still limited under the current setting. Potentially, generations with stronger image-text alignment or training VQA models on a collection of real and synthetic data from scratch can alleviate this.

## 4 CONCLUSION

We've addressed critical gaps in synthetic radiograph generation research by introducing CheX-GenBench, a unified framework to assess generation fidelity, privacy, and clinical utility. Our work highlights key limitations of current models: even state-of-the-art (SoTA) models struggle with long-tailed data distributions, and those with poor fidelity can still pose significant privacy risks. Additionally, while synthetic data shows promise for unimodal tasks, its utility for more complex, multimodal applications remains limited. These observations provide key research directions for future work in synthetic radiograph generation. We envision CheXGenBench to grow with new generative models and paradigms and serve as a dynamic standard for the medical AI community.

In addition to the evaluation framework, our contributions also include the new SoTA model (Sana 0.6B) and SynthCheX-75K, a synthetic dataset comprising 75K high-quality samples with multi-faceted utility. First, it serves as a high-quality standalone training resource, enabling the development of diagnostic models without requiring access to real patient data. Second, it can also be used to augment existing datasets, particularly for rare conditions. Finally, SynthCheX-75K can also be used as a challenging out-of-distribution test set to validate the robustness of new discriminative models trained on real/ synthetic data. **Reproducibility Statement:** To ensure full reproducibility, the code, along with detailed environment setup instructions, is *anonymously* provided here.

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

# A    APPENDIX

# B    BENEFITS OF ADOPTING LLAVA-RAD ANNOTATIONS

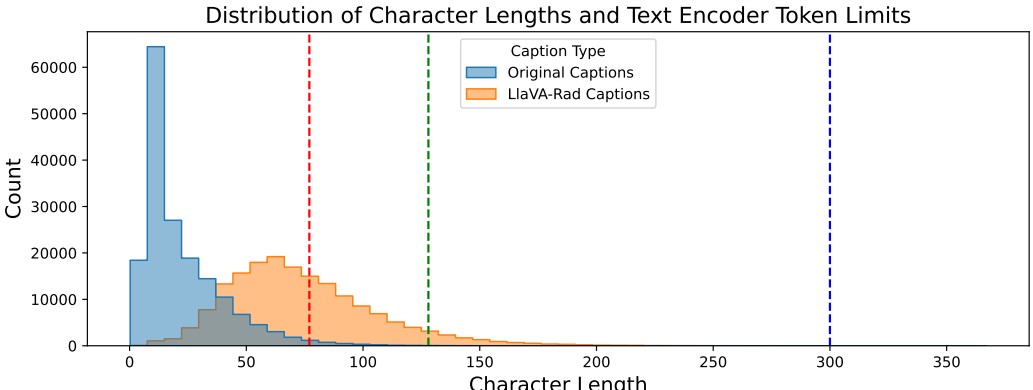

Figure 2: Figure depicting distribution of character lengths for original MIMIC captions and LlaVA-Rad annotations. We also illustrate the text-encoder token length limits for all SD variants and Flux (77 tokens), RadEdit (128 tokens), and Pixart Sigma (300 tokens).
Note: We treat 1 token ≈ 4 characters[2]

Table 6: Comparison of generation fidelity (left) and privacy preservation (right) metrics across different stable diffusion models. LlaVA-Rad annotations consistently outperform original MIMIC impressions, yielding improved image quality (FID/KID (↓), higher alignment (↑)) and enhanced privacy protection (Re-ID (↓), latent/pixel distances (↑)).

| Model | Prompt Type | FID (RadDino) | KID (RadDino) | Alignment Score |
|---|---|---|---|---|
| SD-V1-4 | Original MIMIC | 147.298 | 0.198 | 0.272 |
| SD-V1-4 | LlaVA-Rad | 125.186 | 0.172 | 0.357 |
| SD-V1-5 | Original MIMIC | 144.661 | 0.201 | 0.257 |
| SD-V1-5 | LlaVA-Rad | 118.932 | 0.147 | 0.326 |
| SD-V2 | Original MIMIC | 214.202 | 0.496 | 0.145 |
| SD-V2 | LlaVA-Rad | 194.724 | 0.376 | 0.311 |

(a) Generation fidelity metrics.

| Model | Prompt Type | Max. Re-ID Distance↓ | Min Latent Distance↑ | Min. Pixel Distance ↑ |
|---|---|---|---|---|
| SD-V1-4 | Original MIMIC | 0.725 ± 0.71 | 0.482 ± 0.66 | 132.24 ± 4.6 |
| SD-V1-4 | LlaVA-Rad | 0.539 ± 0.31 | 0.592 ± 0.05 | 143.44 ± 4.6 |
| SD-V1-5 | Original MIMIC | 0.721 ± 0.47 | 0.476 ± 0.41 | 131.44 ± 4.2 |
| SD-V1-5 | LlaVA-Rad | 0.572 ± 0.29 | 0.583 ± 0.04 | 143.634 ± 4.2 |
| SD-V2 | Original MIMIC | 0.687 ± 0.23 | 0.483 ± 0.34 | 132.64 ± 4.3 |
| SD-V2 | LlaVA-Rad | 0.533 ± 0.32 | 0.588 ± 0.05 | 143.936 ± 4.3 |

(b) Privacy and memorisation risk metrics.

In this section, we demonstrate that LLaVA-Rad Annotations lead to substantial improvements in both fidelity performance and reduction of re-identification risks (Table 6). We attribute these improvements to two key factors: the enhanced descriptiveness of the annotations and the removal of certain tokens from the original MIMIC annotations known to increase privacy risks (Dutt, 2025). Figure 2 displays the distribution of average character lengths across both annotation types. MIMIC annotations cluster around significantly smaller values, while LLaVA-Rad Annotations exhibit a wider distribution, indicating greater descriptive detail. Table 6a reveals that LLaVA-Rad Annotations significantly enhance all three fidelity metrics (FID, KID, and Image-Text Alignment) compared to the original MIMIC annotations. Additionally, in Tab. 6b, we observe a substantial improvement in privacy risk mitigation, further validating the superiority of the LLaVA-Rad annotation approach.

# C    UNRELIABILITY IN FIDELITY ESTIMATES WITH OUTDATED BACKBONES

We demonstrate that existing protocols for evaluating generative fidelity in radiographic imaging are unreliable. Current approaches (Bluethgen et al., 2024; Dutt et al., 2024; Lee et al., 2024) calculate image fidelity (FID Score) using an in-domain DenseNet-121 model trained on the MIMIC-CXR

---

[2]https://help.openai.com/en/articles/4936856-what-are-tokens-and-how-to-count-them

Table 7: Comparison of FID and KID metrics across models using two distinct in-domain encoders. While the conventional DenseNet-121 encoder (Cohen et al., 2022) exhibits minimal variance (0.001 - 0.075) across models, indicating limited discriminative power, the RadDino encoder (Pérez-García et al., 2025) demonstrates substantially greater metric differentiation (54.22 - 194.72), providing more meaningful evaluation of model performance.

| | Metric | SD V1-4 | SD V1-5 | SD V2 | SD V2-1 | RadEdit | Pixart Sigma | Sana | SD V3-5 | Lumina 2.0 | Flux.1-Dev | LLM-CXR |
|---|---|---|---|---|---|---|---|---|---|---|---|---|
| **FID** | DenseNet | 0.025 | 0.075 | 0.053 | 0.056 | 0.001 | 0.001 | 0.001 | 0.002 | 0.001 | 0.001 | 0.025 |
| | RadDino | 125.180 | 118.930 | 194.720 | 186.530 | 69.690 | 60.150 | 54.220 | 91.300 | 101.190 | 122.400 | 71.240 |
| **KID** | DenseNet | 0.004 | 0.004 | 0.005 | 0.005 | 0.001 | 0.001 | 0.001 | 0.004 | 0.003 | 0.006 | 0.003 |
| | RadDino | 0.172 | 0.147 | 0.376 | 0.413 | 0.033 | 0.023 | 0.016 | 0.103 | 0.110 | 0.144 | 0.061 |

dataset. We argue this model lacks sufficient discriminative power, resulting in less meaningful fidelity assessments.

In our CheXGenBench benchmark, we address this limitation by leveraging features from RadDino, a state-of-the-art model specifically designed for radiographs. As shown in Table 7, FID evaluation with DenseNet-121 shows minimal variance across models, often ranking several models at the same position. In contrast, the RadDino encoder significantly enhances evaluation quality by providing more meaningful feature representations that better differentiate between model performances.

# D    CORRELATION BETWEEN FIDELITY AND DOWNSTREAM TASKS

**Image Fidelity:** We present the rank for each T2I model across each individual fidelity and mode coverage metric in Tab. 8. We also present the combined rank averaged across all the metrics resulting in Sana (Xie et al., 2025), Pixart Sigma (Chen et al., 2024), and LLM-CXR (Lee et al., 2024) as the top-three performers across all models.

**Downstream Image Classification:** We present the rank for each T2I model across each individual pathology in Tab. 9.

The Pearson correlation between fidelity and classification rank is **0.70**. Based on the correlation coefficient of 0.70 between image fidelity and downstream classification performance, we can derive several important conclusions:

1. **Strong Positive Correlation:** A correlation of 0.70 indicates a strong positive relationship between image fidelity and downstream classification performance. This means that as image fidelity increases, classification performance tends to increase substantially as well. This also supports the value of developing high-fidelity synthetic image generators for medical applications.

2. **Substantial Explained Variance:** The coefficient of determination ($r^2$) would be approximately 0.49, suggesting that about 49% of the variance in classification performance can be explained by image fidelity.

3. **Model Selection Guidance:** When choosing models for generating synthetic medical images for training purposes, prioritizing those with higher fidelity metrics would be a data-driven approach that's likely to yield better downstream performance.

4. **Not a Perfect Relationship:** While strong, the correlation of 0.70 still leaves about 51% of the variance unexplained. This suggests other factors beyond simple image fidelity also influence classification performance, such as:

   (a) Diversity of the generated images
   (b) Representation of edge cases
   (c) Specific features that are diagnostically relevant but might not contribute heavily to overall fidelity metrics

For medical imaging applications specifically, this correlation supports the hypothesis that realistic-looking synthetic images translate to better diagnostic model performance, though the relationship isn't perfect. This finding could help justify investments in more sophisticated image generation techniques that prioritize visual fidelity.

Table 8: Performance ranking of generative models for image fidelity across multiple evaluation metrics (lower rank indicates better performance). The top-3 performers are **(1)** Sana (Xie et al., 2025), **(2)** Pixart Sigma (Chen et al., 2024), and **(3)** LLM-CXR (Lee et al., 2024).

| Model | FID | | KID | | Alignment Score | Precision | Recall | Density | Coverage | Average Rank | Normalized Rank |
|---|---|---|---|---|---|---|---|---|---|---|---|
| | Inception | RadDino | Inception | RadDino | | | | | | | |
| **SD V1-4** | 9 | 9 | 9 | 9 | 4 | 8 | 5 | 7 | 8 | 7.55 | **8** |
| **SD V1-5** | 8 | 7 | 8 | 8 | 5 | 6 | 4 | 6 | 6 | 6.44 | **6** |
| **SD V2** | 10 | 11 | 10 | 10 | 7 | 9 | 6 | 9 | 10 | 9.11 | **10** |
| **SD V2-1** | 11 | 10 | 11 | 11 | 8 | 7 | 7 | 8 | 11 | 9.33 | **11** |
| **RadEdit** | 3 | 3 | 3 | 3 | 3 | 11 | 2 | 10 | 5 | 4.78 | **4** |
| **SD V3-5** | 5 | 5 | 5 | 5 | 10 | 2 | 11 | 4 | 7 | 6.00 | **5** |
| **Lumina 2.0** | 6 | 6 | 6 | 6 | 9 | 5 | 9 | 5 | 9 | 6.78 | **7** |
| **Flux.1-Dev** | 7 | 8 | 7 | 7 | 11 | 10 | 10 | 11 | 4 | 8.33 | **9** |
| **LLM-CXR** | 4 | 4 | 4 | 4 | 6 | 1 | 8 | 1 | 3 | 3.89 | **3** |
| **Pixart Sigma** | 2 | 2 | 2 | 2 | 1 | 4 | 3 | 3 | 2 | 2.33 | **2** |
| **Sana** | 1 | 1 | 1 | 1 | 2 | 3 | 1 | 2 | 1 | 1.44 | **1** |

Note: Lower rank numbers indicate better performance. Top three models highlighted based on normalized rank.

Table 9: Ranking each T2I Model for synthetic data utility (image classification) across all 14 pathologies.

| Model | Atel. | Card. | Cons. | Edema | EC | Fract. | LL | LO | NF | PE | PO | PN | PT | SD | Avg. |
|---|---|---|---|---|---|---|---|---|---|---|---|---|---|---|---|
| SD V1-4 | 5.5 | 5.0 | 6.0 | 5.0 | 7.0 | 5.0 | 2.5 | 4.5 | 5.5 | 5.5 | 6.0 | 7.0 | 5.0 | 3.5 | 5.0 |
| SD V1-5 | 4.0 | 4.0 | 5.0 | 5.0 | 6.0 | 8.5 | 1.0 | 4.5 | 3.0 | 4.0 | 3.5 | 4.0 | 2.5 | 1.5 | 4.0 |
| SD V2 | 7.0 | 6.5 | 7.0 | 7.0 | 3.0 | 8.5 | 7.0 | 7.0 | 7.5 | 7.0 | 10.5 | 5.5 | 6.0 | 5.0 | 7.0 |
| SD V2-1 | 8.0 | 8.0 | 8.0 | 8.0 | 8.0 | 2.0 | 4.0 | 8.0 | 7.5 | 8.0 | 8.5 | 8.0 | 8.0 | 7.0 | 7.0 |
| RadEdit | 3.0 | 2.5 | 1.5 | 2.5 | 3.0 | 7.0 | 6.0 | 2.5 | 4.0 | 2.0 | 2.0 | 1.0 | 4.0 | 6.0 | 3.0 |
| Pixart Sigma | 1.5 | 2.5 | 3.5 | 2.5 | 3.0 | 3.0 | 5.0 | 2.5 | 1.5 | 3.0 | 3.5 | 3.0 | 2.5 | 3.5 | 3.0 |
| **Sana** | **1.5** | **1.0** | **1.5** | **1.0** | **3.0** | **1.0** | **2.5** | **1.0** | **1.5** | **1.0** | **1.0** | **2.0** | **1.0** | **1.5** | **1.0** |
| SD V3-5 | 9.0 | 9.0 | 9.0 | 9.0 | 9.0 | 11.0 | 11.0 | 9.0 | 9.0 | 10.0 | 7.0 | 9.5 | 9.0 | 9.5 | 9.0 |
| Lumina 2.0 | 10.0 | 10.0 | 10.0 | 10.0 | 10.0 | 5.0 | 9.0 | 10.0 | 10.0 | 9.0 | 8.5 | 9.5 | 10.0 | 9.5 | 9.0 |
| Flux.1-Dev | 11.0 | 11.0 | 11.0 | 11.0 | 11.0 | 10.0 | 10.0 | 11.0 | 11.0 | 11.0 | 10.5 | 11.0 | 11.0 | 11.0 | 11.0 |
| LLM-CXR | 5.5 | 6.5 | 3.5 | 5.0 | 3.0 | 5.0 | 8.0 | 6.0 | 5.5 | 5.5 | 5.0 | 5.5 | 7.0 | 8.0 | 6.0 |

Legend: Atel. = Atelectasis, Card. = Cardiomegaly, Cons. = Consolidation, EC = Enlarged Cardiomediastinum, Fract. = Fracture, LL = Lung Lesion, LO = Lung Opacity, NF = No Finding, PE = Pleural Effusion, PO = Pleural Other, PN = Pneumonia, PT = Pneumothorax, SD = Support Devices, Avg. = Average Ranks

# E CORRELATION BETWEEN FIDELITY AND DISEASE DISTRIBUTION

In this section, we examine whether generative fidelity performance for individual pathologies, as reported in Tab. 2, correlates with the frequency of pathology occurrence in the training dataset. Fig. 3 illustrates the occurrence frequency distribution of 14 distinct pathologies in the training set. Conditions such as "No Finding (NF)", "Pleural Effusion (PE)", "Support Devices (SD)", and "Lung Opacity (LO)" represent the most frequently observed pathologies in the training data. Conversely, "Fracture" and "Pleural Other" exhibit significantly lower occurrence frequencies.

Tab. 10 presents the comparative rankings according to occurrence frequency and FID scores. Analysis reveals a remarkably strong positive correlation coefficient of **0.947** between these rankings, providing compelling evidence that generative fidelity demonstrates substantial dependence on pathology occurrence frequency in the training distribution.

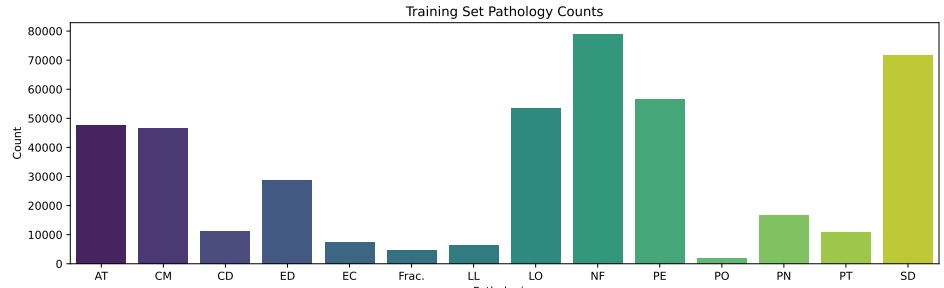

Figure 3: Figure depicting the distribution of pathology counts for the 14 different conditions present in the MIMIC dataset. We indicate pathologies with there abbreviations.
**Note:** **AT** (Atelectasis), **CM** (Cardiomegaly), **CD** (Consolidation), **ED** (Edema), **EC** (Enlarged Cardiomediastinum), **Frac.** (Fracture), **LL** (Lung Lesion), **LO** (Lung Opacity), **NF** (No Finding), **PE** (Pleural Effusion), **PO** (Pleural Other), **PN** (Pneumonia), **PT** (Pneumothorax), **SD** (Support Devices).

Table 10: Occurrence Frequency and Generative Fidelity for Different Pathologies with rankings. We calculate the "Fidelity Rank" across models from Tab. 2

| Pathology Code | Count (n) | Prevalence Rank | FID (RadDino) | Fidelity Rank |
|---|---|---|---|---|
| NF | 78,939 | 1 | 110.75 | 2 |
| SD | 71,537 | 2 | **109.77** | 1 |
| PE | 56,433 | 3 | 130.45 | 5 |
| LO | 53,513 | 4 | 133.77 | 7 |
| AT | 47,704 | 5 | 114.28 | 3 |
| CM | 46,602 | 6 | 114.89 | 4 |
| ED | 28,601 | 7 | 130.75 | 6 |
| PN | 16,832 | 8 | 146.69 | 8 |
| CD | 11,290 | 9 | 172.14 | 9 |
| PT | 10,971 | 10 | 172.15 | 10 |
| EC | 7,454 | 11 | 188.95 | 11 |
| LL | 6,491 | 12 | 194.99 | 12 |
| Frac. | 4,671 | 13 | 211.58 | 13 |
| PO | 2,024 | 14 | 227.43 | 14 |

**Note:** Lower FID (Fréchet Inception Distance) scores indicate better generative fidelity. The table shows a correlation between pathology prevalence and generative quality. Pathology codes: NF = No Finding, SD = Support Devices, PE = Pleural Effusion, LO = Lung Opacity, AT = Atelectasis, CM = Cardiomegaly, ED = Edema, PN = Pneumonia, CD = Consolidation, PT = Pneumothorax, EC = Enlarged Cardiomediastinum, LL = Lung Lesion, Frac. = Fracture, PO = Pleural Other.

## F  EFFECT OF ADDITIONAL FINE-TUNING OF SANA

In this section, we analyze the impact of extended fine-tuning on our benchmark-leading model, Sana (0.6B), by increasing training from 20 epochs (as reported in the main benchmark) to 50 epochs. Our analysis reveals a nuanced picture of how prolonged training affects different performance dimensions. In Tab. 13, we show the improvements from the 20 epoch checkpoint on Report Generation task.

**Fidelity Improvements:** As illustrated in Figure 4a, extended fine-tuning yields modest but consistent improvements in FID scores across all pathologies. The most significant gains were observed in *No Finding*, which stands as the class with the most number of samples in the MIMIC dataset.

**Improvement in Recall Scores:** Fig. 4b demonstrates that recall scores show more substantial improvements than fidelity metrics. This pattern indicates that extended training primarily enhances the model's ability to reproduce a larger spectrum of pathological variations rather than incrementally improving visual quality. Interestingly, in this scenario, all classes (majority and minority) show significant improvement.

**Takeaway:** Despite the additional training epochs, performance improvements on rare pathologies remain disproportionately small compared to common conditions. This observation shows that addressing long-tailed distribution challenges cannot be solved through extended fine-tuning alone and would require specialised algorithmic changes.

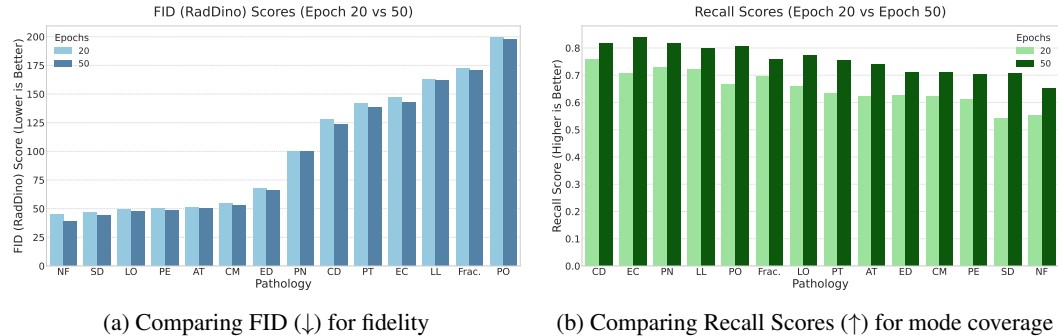

(a) Comparing FID (↓) for fidelity       (b) Comparing Recall Scores (↑) for mode coverage

Figure 4: Extended Training Impact on Sana Model Performance. Comparison of generative quality metrics after standard (20 epochs) and extended (50 epochs) fine-tuning of the Sana model. While FID scores show modest improvement, the Recall metric exhibits substantial enhancement across all pathologies, indicating significantly improved sample diversity without compromising fidelity.

### F.1  TRAINING SETTINGS AND HYPERPARAMETERS

The hyperparameters (learning rates) for Text-to-Image model training are provided in Tab. 11. Existing foundation models (RadEdit, LLM-CXR) were not re-trained and used as is.
For evaluating downstream utility, the learning rates are provided in Tab. 12.

In all Low-Rank Adaptation (LoRA) experiments, the scaling factor $\alpha$ was set to equal the rank (**r**), following the recommended practice. Furthermore, we adopted the rank-stabilized version of LoRA (rsLoRA) (Kalajdzievski, 2023) which has shown to improve convergence during training.

Table 11: Hyperparameters for fine-tuning of T2I models.

| Hyper-Params | SD V1-4 | SD V1-5 | SD V2 | SD V2-1 | RadEdit | Pixart Sigma | Sana | SD V3-5 | Lumina 2.0 | Flux.1-Dev | LLM-CXR |
|---|---|---|---|---|---|---|---|---|---|---|---|
| Fine-Tuning | FFT | FFT | FFT | FFT | N/A | FFT | FFT | LoRA (r-32) | LoRA (r-32) | LoRA (r-32) | N/A |
| Learning Rate | 5e-6 | 5e-6 | 5e-6 | 5e-6 | - | 2e-5 | 1e-4 | 1e-4 | 1e-4 | 1e-4 | - |

Table 12: Hyperparameters for the downstream evaluation tasks.

| Model | ResNet-50 (Classification) | LLaVA-Rad (RRG) |
|---|---|---|
| Fine-Tuning | FFT | LoRA |
| Learning Rate | 1e-4 | 1e-4 |

Table 13: Comparing the performance of fine-tuning Sana from 20 to 50 epochs for the RRG task. Additional fine-tuning does provide benefits, however, they are marginal.

| Model | BLEU-1 | BLEU-4 | ROUGE-L | F1-RadGraph | Micro F1-5 | Micro F1-14 |
|---|---|---|---|---|---|---|
| Original | 38.16 | 15.38 | 0.31 | 0.29 | 0.57 | 0.57 |
| Sana (Epoch 20) | 29.83 | 7.70 | 0.24 | 0.23 | 0.57 | 0.55 |
| Sana (Epoch 50) | 30.80 (+0.97) | 7.91 (+0.21) | 0.25 (+0.01) | 0.24 (+0.01) | 0.58 (+0.01) | 0.57 (+0.02) |

## G  DATA FILTRATION FOR SYNTHCHEX-75K

Generative models can lead to both high and low-fidelity generations on different subsets of the dataset. In order to keep the sample quality high in SynthCheX-75K, a stringent filtration process was adopted using HealthGPT (Lin et al., 2025), a highly-capable medical VLM with advanced understanding, reasoning and generation.

The VLM was provided with the following meta-prompt to classify the quality of each generated sample.

---

**Meta Prompt for HealthGPT**

You are an expert radiologist with extensive experience in medical image interpretation and quality assessment. Your task is to evaluate the quality of a medical image based on its correspondence to the provided clinical description.

Classification Guidelines:

**High Quality:** Excellent image clarity, accurate anatomical representation, clear visibility of described pathological findings, suitable for diagnostic purposes

**Medium Quality:** Acceptable image quality with minor limitations, anatomical structures are recognizable, described findings are visible but may lack optimal clarity

**Low Quality:** Poor image resolution, unclear anatomical structures, difficult to identify described pathological findings, limited diagnostic value

**Not Relevant:** Image content does not correspond to the provided description or shows different anatomical regions/pathologies than described

Response Format:
Provide only one of these four classifications: "High Quality", "Medium Quality", "Low Quality", or "Not Relevant". Do not include explanations or additional commentary.

---

After quality label assignment, images with "Low Quality" and "Not Relevant" labels were removed from the dataset leading to 75,649 samples of high-quality radiographs with pathological annotations. After filtering, SynthCheX-75K constitutes 42,155 (55.72%) "Medium Quality" and 33494 (44.28%) "High Quality" samples, respectively. We have presented the pathology counts by quality in Tab. 14.

## H    Synthetic Medical Data Card for SynthCheX-75K

In accordance with the synthetic medical data (SMD) documentation framework established by (Zamzmi et al., 2024), we present a comprehensive SMD Card for the SynthCheX-75K dataset. This standardized documentation protocol serves multiple critical functions: it systematically captures essential dataset characteristics, including intended applications and scope; provides detailed descriptions of data composition and generation methodology; identifies inherent limitations and potential biases; offers evidence-based recommendations for appropriate usage; and delivers a rigorous assessment of data quality and reliability. The SMD Card framework ensures transparency and reproducibility in synthetic medical data research while facilitating informed decision-making by potential users regarding dataset suitability for their specific applications.

### H.0.1    General Information

---

#### General Information

- **Name:** SynthCheX-75K
- **Release Date:** 14th May 2025
- **Dataset Size:** 137 GB
- **Dataset Modality:** Plain film X-Ray
- **Dataset Provenance:** The synthetic dataset was generated using Sana (Xie et al., 2025), a state-of-the-art text-to-image diffusion model, which underwent domain-specific fine-tuning on medical imaging data. The fine-tuning process involved training the model for 50 epochs using 237,388 chest X-ray samples from the MIMIC-CXR (Johnson et al., 2019) dataset.
- **Dataset Labels:** The dataset contains two types of labels. (1) *multi-label* diagnostic annotations that systematically categorise 14 frequently occurring pathological conditions present in the MIMIC-CXR dataset. (2) Free-text radiology reports providing the clinical findings by a board-certified radiologist.
- Access: The dataset is publicly available on the HuggingFace platform.
- **Licensing:** The dataset is released under the Apache-2.0 license.

---

### H.0.2    Data Quality Evaluation

---

#### Data Quality Evaluation

- **Congruence:** SynthCheX-75K underwent a strict filtration process to remove low-quality generations using a state-of-the-art vision-language assistant (Lin et al., 2025).
- **Coverage:** SynthCheX-75K is derived from MIMIC-CXR and hence, follows a similar distribution in terms of pathologies prevalence, patient race, and demographics.
- **Constraint:** SynthCheX-75K contains only high-quality, clinically-plausible radiographs due to a strict filtration process.
- **Completeness:** The metadata in SynthCheX-75K is complete without any missing values.
- **Compliance:** SynthCheX-75K is derived from the MIMIC-CXR dataset which underwent a de-identification procedure to protect sensitive patient information. However, generative models can generate synthetic images with high patient re-identification risk.

---

### H.0.3 SYNTHETIC DATA USAGE

> **Synthetic Data Usage**
>
> - **Repository Access:** The dataset is publicly available on the HuggingFace platform.
> - **Preprocessing Requirements:** SynthCheX-75K can be directly loaded from HuggingFace without performing any preprocessing steps.
> - **Intended Audience:** SynthCheX-75K is intended to be used researchers and developers as a public synthetic training set for multi-label classification tasks, and (2) a stress-test substrate for evaluating classifiers' robustness to synthetic-only training.

### H.0.4 ETHICAL, LEGAL, AND PRACTICAL CONSIDERATIONS

> **Practical Considerations**
>
> - **Privacy and Anonymization:** SynthCheX-75K is derived from the MIMIC-CXR dataset, which underwent a de-identification procedure to protect sensitive patient information (Protected Health Attributes). However, generative models, especially diffusion models, can generate synthetic images with high patient re-identification risk. Hence, care must be taken while adopting synthetic variants in your use-case.
> - **Biases:** SynthCheX-75K is derived from the MIMIC-CXR dataset, which contains known biases in terms of diagnostic pathology labels (long-tailed disease distribution). Furthermore, MIMIC-CXR was curated at a single institution, leading to potential demographic bias in the patient population.
> - **Recommendations:** SynthCheX-75K is intended to be used researchers and developers as a public synthetic training set for multi-label classification tasks, and (2) a stress-test substrate for evaluating classifiers' robustness to synthetic-only training.

### H.0.5 REFERENCE DATASET GENERAL INFORMATION

> **Reference Dataset General Information**
>
> - **Origin and Source:** The reference dataset for SynthCheX-75K is MIMIC-CXR (Johnson et al., 2019), a large dataset of 227,835 imaging studies for 65,379 patients collected at the Beth Israel Deaconess Medical Centre Emergency Department between 2011–2016.
> - **Dataset Size:** The original uncompressed size of the MIMIC-CXR dataset is 4.6 TB.
> - **Ground Truth Labels:** The dataset contains two types of labels. (1) *multi-label* diagnostic annotations that systematically categorise 14 frequently occurring pathological conditions. (2) Free-text radiology reports providing the clinical findings by a board-certified radiologist.
> - **Metadata:** The dataset contains metadata for View Position (frontal/ lateral), subject ID, study ID, study data and time.
> - **Known Limitations:** The dataset exhibits a long-tailed distribution across the 14 diagnostic pathologies, reflecting the natural prevalence patterns observed in clinical practice. This inherent class imbalance poses substantial challenges, providing a potential source of bias.

## H.1 PATHOLOGY DISTRIBUTION IN SYNTHCHEX-75K

We provide the pathology distribution for SynthCheX-75K in Tab. 14. It can be observed that when sorted in decreasing order in terms of pathology frequency, the distribution in SynthCheX-75K follows that of the original MIMIC-CXR dataset (See Tab. 10).

Table 14: Pathology Counts in SynthCheX and MIMIC Frequency Rank. The pathologies are sorted in decreasing order of prevalence in SynthCheX dataset.

| Pathology (Abbr.) | MIMIC Rank | SynthCheX Count | Medium Quality | High Quality |
|---|---|---|---|---|
| NF (No Finding) | 1 | 24 927 | 6838 | 18 089 |
| SD (Support Devices) | 2 | 23 041 | 14 371 | 8670 |
| PE (Pleural Effusion) | 3 | 18 090 | 13 702 | 4388 |
| LO (Lung Opacity) | 4 | 17 202 | 13 345 | 3857 |
| AT (Atelectasis) | 5 | 15 061 | 11 587 | 3474 |
| CM (Cardiomegaly) | 6 | 14 846 | 10 799 | 4047 |
| ED (Edema) | 7 | 9216 | 7223 | 1993 |
| PN (Pneumonia) | 8 | 5492 | 3920 | 1572 |
| CD (Consolidation) | 9 | 3711 | 2599 | 1112 |
| PT (Pneumothorax) | 10 | 3602 | 2281 | 1321 |
| EC (Enlarged Cardiomediastinum) | 11 | 2352 | 1634 | 718 |
| LL (Lung Lesion) | 12 | 2099 | 1437 | 662 |
| Frac. (Fracture) | 13 | 1474 | 938 | 536 |
| PO (Pleural Other) | 14 | 655 | 442 | 213 |

# I  ABLATIONS OF LoRA

**Ablations on Rank:** Our ablation study on the LoRA rank for large models (>1B parameters), presented in Tab. 15, reveals that increasing the rank modestly improves generation fidelity. However, our key finding is that a fully fine-tuned smaller model (Sana, 0.6B) still significantly outperforms much larger models adapted with PEFT (e.g., SD V3.5, 2.5B; Flux.1-Dev, 1.2B). This result is crucial for the medical image analysis community, as it highlights that thorough adaptation of an efficient model can be more effective and accessible than resource-intensive scaling of larger architectures. For each experiment, the scaling factor alpha ($\alpha$) was set equal to the rank following the recommended practice.

Table 15: Performance comparison across different generative models and LoRA ranks. FID scores calculated using RadDino metric (lower is better). Best performance for each model is highlighted in bold.

| Model | FID Score (RadDino) by LoRA Rank | | |
|---|---|---|---|
| | Rank 32 | Rank 64 | Rank 128 |
| SD v3.5 | 91.30 | 84.14 | **74.58** |
| Lumina 2.0 | 101.19 | 96.51 | **88.28** |
| Flux.1-Dev | 122.40 | 105.28 | **95.17** |
| Average | 104.96 | 95.31 | **85.68** |

**Ablations on LoRA Position:** We conducted additional ablation studies on the placement of LoRA modules on SD v3.5. Specifically, we used a fixed rank ($\mathbf{r}$) and alpha ($\alpha$) of 32 and explored two additional positions beyond the standard Attention Layers: **(1)** Attention + MLP layers and **(2)** Attention + MLP + Positional Embedding. The results are presented in Tab. 16.
The ablation study indicates that, for the fixed rank $\mathbf{r} = \mathbf{32}$, none of the tested LoRA placement strategies yielded a significant improvement over the baseline placement in the Attention layers alone. While integrating LoRA into both the Attention and MLP layers showed a minor improvement, including the Positional Embedding layer resulted in a notable degradation of performance. These results suggest that applying LoRA to the Positional Embedding may disrupt vital spatial information. Future work should investigate whether substantial gains can be achieved by applying the most promising configurations (Attention + MLP) with a higher LoRA rank (e.g., $r = 128$)

Table 16: Impact of LoRA Position on FID Score (r=32, alpha=32)

| Metric/Layer (Fixed r=32) | Attention Only | Attention + MLP | All Layers (Attn + MLP + Pos Emb.) |
|---|---|---|---|
| FID (RadDino) | 91.30 | 85.26 | 99.19 |

## J  DOWNSTREAM UTILITY ON IMAGE SEGMENTATION

We extend our experimental framework for evaluating synthetic data utility by incorporating an image segmentation task, complementing the established (1) Image classification and (2) Radiology Report Generation (RRG) tasks (Sec. 3.3).

**Methodology:** Since the MIMIC-CXR dataset lacks pixel-level annotations, we established a surrogate task of segmenting the clavicles. To generate the necessary supervision, we utilized the segmentation model proposed by Seibold et al. (2023), which is capable of delineating 159 distinct anatomical regions. This process yielded *pseudo* ground-truth masks, absent in the original dataset. Subsequently, we trained a U-Net architecture (Ronneberger et al., 2015) using these image-mask pairs. To assess whether synthetic data provides performance benefits for segmentation, we conducted controlled experiments by augmenting the real training data with synthetic samples.

**Experimental Setting:** For the real-data baseline, we trained the U-Net on 3,000 real images from MIMIC-CXR. For the augmentation experiments, we supplemented this baseline with 3,000 synthetic images, resulting in a combined training set of 6,000 samples. All models were trained for 15 epochs with a learning rate of 5e-5. The checkpoint achieving the best performance on the validation split was selected for final inference. To ensure consistency across the benchmark, the segmentation evaluation employs the same held-out test set of 5,034 samples used for the classification task. The results are presented in Tab. 17.

| Method | Augmentation Source | Size | DICE |
|---|---|---|---|
| **Baseline** | *None* | 3,000 | 0.669 |
| *Augmented Models* | | | |
| | SD V1-4 | 6,000 | 0.593 |
| | SD V1-5 | 6,000 | 0.586 |
| | SD V2 | 6,000 | 0.537 |
| | SD V2-1 | 6,000 | 0.541 |
| | RadEdit | 6,000 | 0.667 |
| | Pixart $\Sigma$ | 6,000 | **0.671** |
| | Sana | 6,000 | 0.667 |
| | SD V3-5 | 6,000 | 0.612 |
| | Lumina 2 | 6,000 | 0.581 |
| | Flux.1 | 6,000 | 0.562 |
| | LLM-CXR | 6,000 | 0.642 |

Table 17: Performance comparison of baseline and augmented models.

**Results:** The utility of synthetic augmentation for segmentation exhibits a strong dependency on generative fidelity, revealing a clear divergence between model architectures (Table 17). We observe that **high-fidelity models** (e.g., Sana, Pixart $\Sigma$) are capable of matching or marginally surpassing the real-data baseline (Best Augmented: **0.671** vs. Baseline: 0.669), effectively demonstrating that state-of-the-art generative models can preserve the structural integrity required for pixel-level tasks. In contrast, augmenting with data from **low-fidelity models** (e.g., SD V2, Flux.1) led to significant performance degradation (e.g., SD V2 dropped to 0.537). We attribute this disparity to the "Texture-Topology Gap". While the high-fidelity models are able to preserve the texture of the real radiographs, the low-fidelity models further worsen it, leading to performance degradation. Segmentation U-Nets rely on high-frequency edge gradients, specifically the noisy transitions between bone and soft tissue found in real radiographs. Low-fidelity synthetic images often exhibit a "waxy" or denoised texture

with artificially smooth gradients, creating a domain shift that confuses the U-Net. However, the success of models like Pixart Σ and Sana suggests that recent advancements in diffusion transformers are beginning to bridge this gap, capturing sufficient microscopic texture to serve as viable training data for segmentation, whereas earlier architectures fail to generalize. In Appendix L, we further analyse and compare the structural differences between real and synthetic radiographs.

Finally, we highlight that our segmentation evaluation is constrained by the inherent lack of pixel-level annotations within the MIMIC-CXR dataset. To address this, we adopted a distillation-based approach, utilizing an established external segmentation model (Seibold et al., 2023) to generate pseudo-labels (ground truth masks) for the training data. Consequently, the downstream performance reported here is strictly conditioned on and effectively upper-bounded by the precision of this "oracle" model. We hypothesize that while an ideal experimental design would utilize radiologist-verified contours to eliminate label noise, this proxy method allows for a relative assessment of synthetic data utility. Furthermore, our current evaluation focuses on rigid anatomical structures, specifically the Clavicles. While these tasks provide a standardized baseline for assessing structural coherence, they represent a simplified proxy for clinical utility. A more rigorous evaluation of generative fidelity would be pathology segmentation (e.g., delineating the precise boundaries of a Pneumothorax or Pleural Effusion), where the model must distinguish dynamic pathological textures from the underlying healthy anatomy. We defer these extensions to future studies.

## K  EXPERIMENTS ON SYNTHETIC DATA AUGMENTATION

In Section 3.3 (Tables 4 and 5), we evaluated the standalone utility of synthetic data by training models exclusively on generated images (the synthetic-only regime). This established the baseline efficacy of generative models as independent data sources. In this section, we extend our analysis to the synthetic augmentation regime, assessing the complementary value of synthetic data when integrated with real-world clinical datasets. Specifically, we evaluate performance gains in downstream image classification and radiology report generation when a real image dataset is expanded with synthetic examples, thereby testing the hypothesis that generative models can benefit model training by augmenting the information contained in a real data distribution.

### K.1  IMAGE CLASSIFICATION

Table 18: **Downstream Classification Utility (AUC Scores).** Comparison of baseline (Real Data) against Synthetic Augmentation. The final column represents the row-wise average AUC. Instances where performance gain for minority classes (Fracture, PO) is observed are highlighted .

| Model | Atelec. | C.Megaly | Consol. | Edema | Enlarged C. | Frac. | LL | LO | NF | PE | PO | PN | PT | SD | Average |
|---|---|---|---|---|---|---|---|---|---|---|---|---|---|---|---|
| Original | 0.75 | 0.76 | 0.72 | 0.85 | 0.61 | 0.58 | 0.63 | 0.70 | 0.84 | 0.84 | 0.74 | 0.67 | 0.71 | 0.83 | 0.73 |
| SD V1-4 | 0.76 | 0.78 | 0.74 | 0.87 | 0.65 | 0.60 | 0.73 | 0.73 | 0.85 | 0.85 | 0.74 | 0.70 | 0.77 | 0.87 | **0.76** |
| SD V1-5 | 0.76 | 0.77 | 0.74 | 0.88 | 0.66 | 0.60 | 0.72 | 0.73 | 0.85 | 0.86 | 0.72 | 0.71 | 0.77 | 0.87 | **0.76** |
| SD V2 | 0.76 | 0.78 | 0.73 | 0.87 | 0.64 | 0.58 | 0.69 | 0.72 | 0.85 | 0.86 | 0.70 | 0.69 | 0.76 | 0.86 | **0.75** |
| SD V2-1 | 0.76 | 0.77 | 0.73 | 0.87 | 0.64 | 0.58 | 0.68 | 0.72 | 0.85 | 0.85 | 0.72 | 0.69 | 0.75 | 0.86 | **0.75** |
| RadEdit | 0.76 | 0.77 | 0.75 | 0.87 | 0.66 | 0.62 | 0.70 | 0.72 | 0.85 | 0.86 | 0.78 | 0.69 | 0.77 | 0.85 | **0.76** |
| Pixart Σ | 0.76 | 0.78 | 0.75 | 0.87 | 0.66 | 0.62 | 0.73 | 0.73 | 0.85 | 0.85 | 0.75 | 0.70 | 0.78 | 0.86 | **0.76** |
| Sana | 0.76 | 0.78 | 0.74 | 0.87 | 0.66 | 0.63 | 0.70 | 0.73 | 0.85 | 0.86 | 0.77 | 0.69 | 0.78 | 0.86 | **0.76** |
| SD V3-5 | 0.74 | 0.76 | 0.72 | 0.85 | 0.63 | 0.61 | 0.65 | 0.70 | 0.84 | 0.84 | 0.70 | 0.66 | 0.72 | 0.84 | 0.73 |
| Lumina 2.0 | 0.74 | 0.74 | 0.73 | 0.85 | 0.63 | 0.62 | 0.65 | 0.71 | 0.84 | 0.84 | 0.65 | 0.65 | 0.70 | 0.84 | 0.73 |
| Flux.1-Dev | 0.75 | 0.72 | 0.72 | 0.84 | 0.61 | 0.61 | 0.66 | 0.69 | 0.84 | 0.83 | 0.69 | 0.66 | 0.68 | 0.83 | 0.73 |
| LLM-CXR | 0.76 | 0.76 | 0.74 | 0.87 | 0.66 | 0.59 | 0.69 | 0.70 | 0.82 | 0.83 | 0.74 | 0.68 | 0.75 | 0.85 | **0.75** |

**Experimental Setup:**  To evaluate the impact of synthetic data augmentation, we constructed composite training sets by augmenting the baseline dataset of 20,000 real radiographs with an equal volume (1:1 ratio) of synthetic samples generated by each candidate text-to-image model. Crucially, the synthetic samples were conditioned on the original radiology reports corresponding to the real training subset to ensure semantic alignment. For downstream classification, we fine-tuned an ImageNet-pretrained ResNet-50 architecture. Training was conducted for 20 epochs using the Adam optimizer with a learning rate of $1 \times 10^{-4}$ and a global batch size of 768, distributed across four NVIDIA A100 GPUs. Model selection was performed based on the lowest loss on a held-out validation set, and the best-performing checkpoint was subsequently evaluated on the established test

set ($N = 5,034$). We benchmark our results against the baseline performance of a model trained exclusively on the 20,000 real samples, as established in Table 4.

**Results:** The results show an interesting phenomenon. On average, data from each T2I model (irrespective of the generative fidelity) either matches or outperforms the real data baseline. For instance, SD V1-4 (FID ≈ 125) performs equally well as Sana (FID ≈ 54), leading to a 3% improvement in the average AUC. Furthermore, minority pathologies such as *Pleural Other*, where most T2I models demonstrated low FID scores, outperform the real data baseline on several instances (RadEdit, Pixart-Σ, Sana). For most prevalent pathologies such as Atelectasis (AT), Cardiomegaly (CM), and Support Devices (SD), most T2I models either match or outperform the real data baseline.

We attribute these surprising results to the **Texture-Topology Gap** (further explained in Appendix L). Specifically, we find that T2I models generate synthetic X-rays with *sufficient semantic correctness* (macroscopic topology), i.e. correct placement of the heart, ribs, lungs, and other organs. This semantic correctness is utilized by the downstream classifier (ResNet50) for better generalization, which leads to higher AUC scores. However, the same models fail to reproduce the microscopic texture, i.e. granularity and contrast of bones, often synthesizing a smooth, "waxy" appearance in the synthetic X-rays (see Fig. 5). This inability to reproduce the microscopic texture degrades the FID scores. In Appendix J, we discuss that the discrepency in texture consistency can adversely affect more fine-grained tasks like segmentation. On the contrary, our results suggest that for classification tasks, the downstream model is not reliant on subtle texture details and can can learn from the synthetic images' macroscopic anatomy that the diffusion models generate well. Furthermore, the comparable performance gains observed across models with vastly different FID scores support the hypothesis of a **Utility Plateau**. Once a generative model surpasses a certain degree of semantic correctness (correct organ and pathology placement), further improvements in texture may yield diminishing returns for discriminative classification. These findings provide an avenue for future studies to explore this phenomenon in detail.

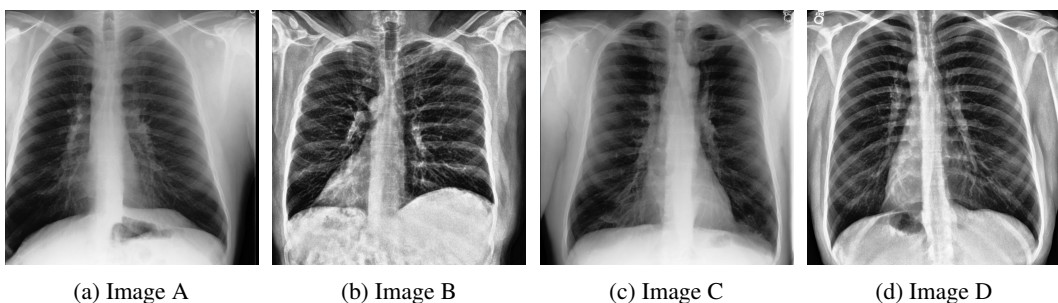

|     |     |     |     |
| --- | --- | --- | --- |
| (a) Image A | (b) Image B | (c) Image C | (d) Image D |

Figure 5: Example of synthetic generations from SD V2 (FID ≈ 194). While the generations show an inconsistent texture (enhanced contrast, "waxy" appearance of bones), the semantic consistency (organ placement) is maintained.

## K.2 RADIOLOGY REPORT GENERATION

**Training Protocol:** LLaVA-Rad (Zambrano Chaves et al., 2025) was originally pretrained on a massive corpus of 697,000 image-text pairs derived from diverse public and private sources. Replicating this extensive pretraining regime presents significant barriers in terms of data accessibility and computational resources. Consequently, we adopted a streamlined training protocol designed to isolate the impact of synthetic augmentation. We constructed a baseline training set using a stratified subset of 50,000 real samples from MIMIC-CXR, preserving the original prevalence ratios of all pathology classes. To evaluate the utility of generative augmentation, we supplemented this baseline with synthetic samples generated by each candidate text-to-image model, allowing for a direct assessment of how synthetic data quality influences report generation performance relative to a controlled real-data baseline.

**Experimental Setup:** LLaVA-Rad utilizes a two-stage training pipeline, following the original LLaVA protocol (Liu et al., 2023). In the first stage, the visual encoder is frozen, and only the MLP projector is trained to align visual features with the language model's embedding space. In the

second stage, the projector and the LLM are jointly fine-tuned, with the LLM parameters updated via Low-Rank Adaptation (LoRA) (Hu et al., 2022).

In contrast to our *synthetic-only* experiments, where we performed continual fine-tuning on a pre-trained checkpoint using only synthetic data (Stage 2), our augmentation protocol involves training the entire model from scratch through both stages (*train-from-scratch* setting). This ensures that the foundational feature alignment (Stage 1) learns from the combined distribution of real and synthetic data. For both stages, we adhered to the standard LLaVA-Rad hyperparameters, utilizing learning rates of $1 \times 10^{-3}$ (Stage 1) and $1 \times 10^{-4}$ (Stage 2) over 10 epochs. All other optimization settings remain consistent with the original LLaVA-Rad implementation. Evaluation was conducted using the same comprehensive metric suite previously adopted for RRG tasks.

**Objectives:** Our experimental design for this phase was driven by two key research questions. First, we sought to determine whether synthetic data augmentation yields performance gains when training a multimodal model from scratch (encompassing both Stage 1 feature alignment and Stage 2 low-rank fine-tuning), in contrast to the continual fine-tuning regime explored in Section 3.3. Second, we aimed to observe if gains in generative fidelity translate to downstream RRG performance, assessing whether higher perceptual quality in synthetic training data translates to more precise diagnostic text generation.

Table 19: Performance comparison of LLaVA-Rad trained from scratch with real data only versus real data augmented with synthetic images from various text-to-image models and the SyntheCheX-75K dataset.

| Metric | Original (Real Data Only) (N=50,000) | Real Data + SD V1-4 (N=70,000) | Real Data + SD V1-5 (N=70,000) | Real Data + RadEdit (N=70,000) | Real Data + Pixart (N=70,000) | Real Data + Sana (N=70,000) | Real Data + SyntheCheX (N=125,000) |
|---|---|---|---|---|---|---|---|
| BLEU-1 | 27.26 | 24.49 | 25.62 | 31.40 | 32.53 | 32.83 | **33.90** |
| BLEU-4 | 10.29 | 8.76 | 8.95 | 10.49 | 11.26 | 11.28 | **12.31** |
| ROUGE-L | 0.23 | 0.20 | 0.22 | 0.26 | 0.26 | 0.26 | **0.28** |
| F1-RadGraph | 0.21 | 0.19 | 0.22 | 0.26 | 0.26 | 0.26 | **0.27** |
| Micro F1-5 | 0.45 | 0.41 | 0.41 | 0.49 | 0.51 | 0.52 | **0.54** |
| Micro F1-14 | 0.45 | 0.39 | 0.40 | 0.50 | 0.52 | 0.52 | **0.55** |
| GREENScore | 0.27 | 0.27 | 0.27 | 0.31 | 0.31 | 0.32 | **0.34** |
| RaTEScore | 0.45 | 0.49 | 0.49 | 0.50 | 0.52 | 0.53 | **0.55** |

**Results:** The results in Table 19 reveal a critical divergence from the trends observed in image classification. Unlike the classification task, where low-fidelity models (e.g., SD V1-5) also provided performance gains, the RRG task demonstrates a strong dependency on generative fidelity. Augmenting with low-fidelity synthetic data (SD V1-4, SD V1-5) resulted in a notable degradation of performance across most metrics. For instance, SD V1-4 dropped the BLEU-4 score from the baseline 10.29 to 8.76, and F1-RadGraph from 0.21 to 0.19. This confirms our hypothesis that RRG is a texture-sensitive task. The "waxy" or denoised artifacts in low-fidelity images likely confuse the visual encoder, leading to hallucinations or generic descriptions that fail to match the granular clinical ground truth. Conversely, augmenting with high-fidelity models (RadEdit, Pixart, Sana) yielded substantial improvements. Sana achieved a BLEU-4 of 11.28 and a RaTEScore of 0.53 (vs. baseline 0.45) showcasing that the RRG task benefit when both macroscopic topology and microscopic texture are well-captured in the synthetic images. Finally, the SyntheCheX-75K dataset provided the highest gains (BLEU-4: 12.31, RaTEScore: 0.55), suggesting that scaling up high-quality synthetic data further amplifies this benefit.

Comparing the results of our Training from Scratch (Stage 1 and 2) experiment above versus Continual Fine-Tuning (Stage 2 only) in Section 3.3, we see that synthetic data delivers consistent gains in the former condition but worsens performance in the latter. We attribute the latter result to the small residual divergence between real and synthetic data, causing catastrophic forgetting (of knowledge learned from the real data) when used to fine-tune the original Llava-Rad. E.g., The model may specialise too much to the waxy textures of the synthetic data, and then perform poorly for the more textured real test data. Meanwhile, using both real and synthetic data together allows us to benefit from the additional synthetic data without forgetting the original real data.

## L    A DEEPER ANALYSIS OF SYNTHETIC DATA SEMANTICS

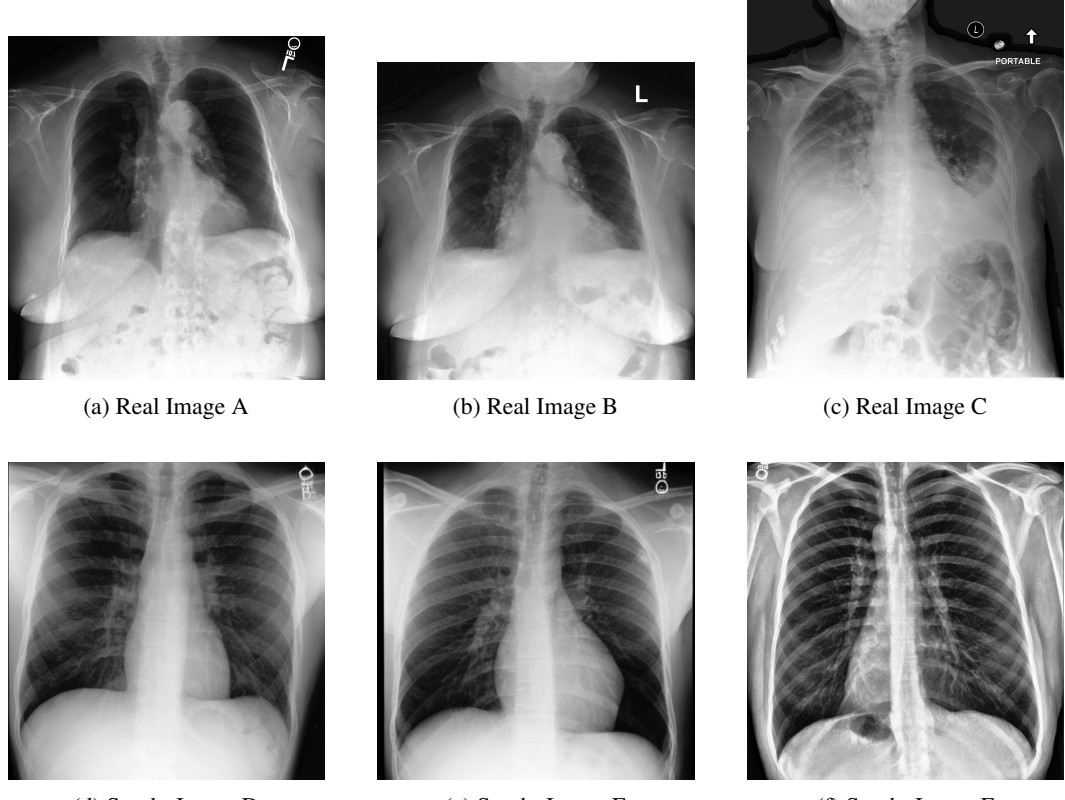

| (a) Real Image A | (b) Real Image B | (c) Real Image C |
| (d) Synth. Image D | (e) Synth. Image E | (f) Synth. Image F |

Figure 6: Comparison of texture and appearance between real and synthetic radiographs. Top row: Real data; Bottom row: Synthetic data.

In this section, we dive deeper into the semantics of the synthetic images and study how they compare against the real images. This analysis would enable us to understand where and why synthetic data augmentation provides performance gains.

In Fig. 6, we visually compare different examples of real (top row) and synthetic radiographs (bottom row). The figure visually substantiates the divergence between semantic coherence and perceptual fidelity. As observed in the real radiographs (Top Row), clinical data is characterized by high-frequency noise profiles (often arising from data-acquisition protocols), intricate vascular branching, and distinct trabecular bone texture (intricate mesh-like structure that gives a grainy appearance to bones). In contrast, the synthetic generations (Bottom Row) exhibit a "waxy" or "hyper-smooth" appearance. This can be observed with a simple visual inspection of the synthetic images, where the ribs appear as if they have been "hand-drawn" and lack internal granularity. For instance, the spine (central vertical bone) in real images showcases distinct vertebrae (small columns) while they appear as a single, vertical block in synthetic images, indicating a lack of granular details. This observation also explains the performance degradation observed with synthetic data augmentation discussed in Appendix J.

**The Texture-Topology Gap:** We explain the performance difference with and without synthetic data augmentation through the Texture-Topology Gap. On a macroscopic level, the *topology* of the radiograph represents the geometric arrangement of anatomical structures, such as the cardiac silhouette, rib cage curvature, and large pathological opacities. Our visual analysis confirms that the T2I models preserve this signal with high fidelity, correctly placing organs and lesions in plausible anatomical locations. This explains the performance gains observed in the case of image classification and RRG, which rely on the high-level topology of the X-ray. However, pixel-level tasks such as image segmentation rely on the *microscopic-texture* such as bone granularity, pulmonary vascular

branching and inherent noise due to data acquisition protocols. Synthetic generations consistently fail to reproduce this signal, instead exhibiting a smoothed, "waxy" appearance (Fig. 6) that creates a domain shift and provides minimal/ no performance improvements.

## M   FORMALISATION OF PRIVACY METRICS

Given a dataset of $M$ generated images $\{\hat{x}^{(j)}\}_{j=1}^{M}$ (across $M$ different random seeds), let $s_{\text{reid}}^{(j)}, \ell_{\text{lat}}^{(j)}$, and $\ell_{\text{pix}}^{(j)}$ denote, respectively, the Re-ID score, latent-space distance, and pixel-space distance of sample $j$ to its closest training image. We report the following dataset–level statistics:

$$\textbf{Avg. Re-ID Score } (\downarrow) : \overline{s}_{\text{reid}} = \frac{1}{M} \sum_{j=1}^{M} s_{\text{reid}}^{(j)}, \qquad \textbf{Avg. Latent Distance } (\uparrow) : \overline{\ell}_{\text{lat}} = \frac{1}{M} \sum_{j=1}^{M} \ell_{\text{lat}}^{(j)},$$

$$\textbf{Avg. Pixel Distance } (\uparrow) : \overline{\ell}_{\text{pix}} = \frac{1}{M} \sum_{j=1}^{M} \ell_{\text{pix}}^{(j)}, \qquad \textbf{Max. Re-ID Score } (\downarrow) : s_{\text{reid}}^{\max} = \max_{1 \leq j \leq M} s_{\text{reid}}^{(j)},$$

$$\textbf{Count}[s_{\text{reid}} > \delta] \, (\downarrow) : C_{\delta} = \sum_{j=1}^{M} \mathbf{1}\big[s_{\text{reid}}^{(j)} > \delta\big].$$

