# OpenReview forum: "CheXGenBench: A Unified Benchmark for Fidelity, Privacy and Utility of Synthetic Chest Radiographs"
_ICLR.cc/2026/Conference — Submitted to ICLR 2026_

### Official Review · Reviewer_nMkj · 2025-10-25

**Soundness:** 3
**Presentation:** 3
**Contribution:** 3
**Rating:** 8
**Confidence:** 1

**Summary:**

The paper presents a comprehensive evaluation framework for T2I model performance in medical imaging. It addresses a clear gap in the field: the absence of standardized, reproducible benchmarks for synthetic radiograph generation. CheXGenBench evaluates models along three complementary dimensions-fidelity, privacy, and utility-using over twenty quantitative metrics applied to eleven leading generative architectures, ranging from Stable Diffusion variants to modern transformer-based models.

The benchmark standardizes training and evaluation protocols, enabling equitable model comparison. It integrates domain-adapted encoders such as RadDino and BioViL-T for in-domain evaluation, improving over prior works that relied on natural-image metrics like Inception-based FID. Moreover, it introduces pathology-conditional evaluation, offering fine-grained insights into model performance on long-tailed medical distributions. The framework also incorporates privacy risk quantification through pixel, latent, and re-identification distance metrics, reflecting growing regulatory and ethical concerns in synthetic medical imaging.

Beyond benchmarking, the authors release SynthCheX-75K, a synthetic dataset of 75000 chest radiographs generated by the top-performing model (Sana 0.6B), intended for downstream diagnostic research and data augmentation. The results show that while synthetic data can match real data performance in unimodal classification tasks, multimodal utility, such as report generation, remains limited.

The paper is well-structured, clearly written, and timely, offering a valuable contribution toward standardization in synthetic medical imaging. A minor limitation is that the benchmark, while broad, remains confined to chest X-rays and might not generalize to other modalities. Additionally, some metrics overlap conceptually, and further justification for their selection would strengthen clarity. Nonetheless, the unified design, release of reproducible code, and transparent reporting mark an important step forward for evaluating generative models in healthcare.

**Strengths:**

- Original contribution introducing the first unified benchmark combining fidelity, privacy, and clinical utility for synthetic radiograph generation.
- Addresses a clear and relevant gap in medical AI benchmarking with a well-motivated framework.
- High methodological quality with standardized training and evaluation protocols ensuring fair model comparison.
- Integration of in-domain models (RadDino, BioViL-T) improves validity of generative fidelity assessment.
- Novel pathology-conditional evaluation provides granular insights into model behavior on long-tailed medical distributions.
- Inclusion of privacy and re-identification risk metrics is timely and important for ethical AI in healthcare.
- Clear presentation, consistent structure, and well-organized figures and tables aid readability.
- Release of SynthCheX-75K dataset adds practical value and supports community reproducibility.
- Significance lies in establishing a reproducible, extensible benchmark that can evolve with future generative model advances.
- Balanced discussion of current model limitations and future research directions demonstrates maturity and impact.

**Weaknesses:**

- Benchmark scope is limited to chest X-rays, reducing generalizability to other medical imaging modalities such as CT, MRI, or ultrasound.
- The choice and weighting of some metrics are not fully justified, and the overlap between fidelity measures (FID, KID, PRDC) may obscure interpretability.
- Privacy evaluation lacks ground truth for real-world re-identification risk, relying solely on similarity metrics without human or institutional validation.
- Downstream utility experiments cover only classification and report generation, leaving out segmentation and localization tasks that are clinically relevant.
- While standardized training is valuable, fixed hyperparameters across diverse architectures may disadvantage some models and affect fairness.
- Synthetic data quality is discussed mainly quantitatively; more qualitative visual examples or expert reader assessments could strengthen conclusions.
- Limited discussion on how the benchmark will be maintained or expanded, which may affect long-term community adoption.

**Questions:**

1. How easily can CheXGenBench be extended to other modalities such as CT or MRI, and are there plans to do so?
 2. How were the relative weights or importance of the 20+ metrics decided, and would metric correlation analysis help simplify interpretation?
 3. For privacy evaluation, have any real re-identification or membership inference experiments been performed to validate metric reliability?
 4. Could the benchmark include additional downstream tasks like segmentation or detection to better reflect clinical utility?
 5. Were hyperparameters adjusted per model, or were all architectures evaluated under identical training settings?
 6. What quality control or expert review was applied to SynthCheX-75K to ensure diagnostic plausibility?
 7. Are there plans for regular updates or a public leaderboard to keep the benchmark relevant as new models emerge?

---

> ### Author Response · Authors · 2025-11-27
> **Author Response (1/3)**
>
> **[1] Limited Scope, Generalizability to other medical imaging modalities**
> We thank the reviewer for raising this important point. While the benchmark is designed to be easily extensible and generalizable to other modalities (CT, MRI, Ultrasound), our initial focus on chest X-rays was a deliberate strategic decision due to the following reasons:
>    - **Most active area of research:** In Section 2.1, we state that the MIMIC-CXR dataset and the Chest X-Ray modality have become the **"de-facto standard"** for text-to-image generation of medical images. The field is highly active, with numerous studies adopting this dataset (Bluethgen et al., 2024; Weber et al., 2023; Pérez-García et al., 2024; Lee et al., 2024; Dutt et al., 2024). However, this research has been fragmented, with "inconsistent evaluation protocols", “fragmented assessment criteria” and a lack of comparative baselines. Our primary goal was to first bring order to this specific, active domain, allowing for the rigorous, fair comparisons that uncovered our key findings.
>    - **Generalizability to Other Modalities:** While the current implementation is domain-specific, the benchmark's core design is modular and modality-agnostic. The principles of evaluating Fidelity, Privacy, and Utility are universal. **As stated in Section 2**, CheXGenBench was deliberately designed to have decoupled training and evaluation pipelines. This ensures that the benchmark can be easily extended to other modalities.
>
>       - In order to adapt CheXGenBench to other modalities (CT, MRI, etc), a researcher only needs to make the following two modifications:
>          - Train Text-to-Image (T2I) models on a modality-specific dataset
>          - Swapping our domain-specific image encoder (RadDino for CXR) with an appropriate SoTA encoder for their domain (e.g., a 3D CT-specific encoder).
>          - The remainder of the process for obtaining fidelity, utility and privacy metrics would remain exactly the same.
>
> **While extending to other modalities is straightforward in terms of the training protocol**, it would demand substantial computational resources for the training and evaluation of numerous Text-to-Image models. Therefore, we leave this a promising avenue for future work.
>
> **[2] How were the relative weights or importance of the 20+ metrics decided?**
> Thank you for this insightful question regarding metric selection and interpretability.
>
> Our goal was to create a comprehensive multi-metric "scorecard" rather than combining metrics into a single, arbitrarily weighted score, as different applications may prioritize fidelity, privacy, or utility differently.  We address the concerns below:
>
>    - **Justification for a Metric Suite:** There was no importance weighting. We intentionally did not assign specific weights to the 20+ metrics to create a single aggregate score.
> Our framework is designed to provide a multi-dimensional, comprehensive assessment, allowing researchers to prioritize metrics based on their specific goals. For instance, a user focused on data augmentation for rare diseases would prioritize Recall and pathology-specific FID, while a user deploying a model for general use would be highly concerned with Re-ID Scores. However, we do present combined ranks for each Text-to-Image model based on the aggregate ranks across all fidelity metrics in **Appendix D, Table 8.**
>
>    - **Would metric correlation analysis help simplify interpretation?** We agree that a correlation analysis is crucial for interpretation, and we included this analysis in our paper. In Appendix D, we explicitly conduct a correlation analysis between our generative fidelity metrics and the downstream utility task of image classification. This analysis is critical as it validates that our fidelity metrics are meaningful (showing a strong 0.70 correlation with task performance). This analysis revealed that when choosing a Text-to-Image model, prioritizing those with higher fidelity metrics is likely to yield better downstream performance. However, it also demonstrates that fidelity is not a perfect proxy for utility, thus justifying our benchmark's multi-faceted evaluation of both.

---

> ### Author Response · Authors · 2025-11-27
> **Author Response (2/3)**
>
> **[3] Concerns about Privacy Evaluations**
> It was not straightforwardly possible to conduct real-world re-identification attacks, given the original dataset’s anonymity and the need to treat data and patients ethically. We confirm that our evaluation protocol explicitly includes a rigorous, deep learning-based re-identification experiment.
>    - **Deep Learning-Based Re-Identification Attack:** Our primary privacy metric utilizes a Siamese neural network $f_{\theta}^{re-id}$ with a ResNet-50 backbone, trained specifically to classify whether two radiographs originate from the same patient. This serves as a "real" and potent inference attack, quantifying the probability $s_{reid}$ that a generated image leaks the biometric identity of a specific patient from the training set.
>    - **Alignment with Established Standards:** This methodology is the established standard for quantifying privacy leakage in medical imaging literature (Fernandez et al., 2023; Dutt et al., 2025; Dutt, 2025). By adopting this metric alongside complementary pixel and latent distance measures, we ensure our benchmark unifies existing studies while providing a robust, multi-faceted assessment of re-identification risk.
>
> **[4] Additional Downstream Tasks**
> We thank the reviewers for suggesting the inclusion of an image segmentation task. We agree that this addition provides a critical perspective on the pixel-level utility of synthetic data. We have updated the manuscript to include a **new experimental section in Appendix J**, detailing the methodology, protocol, and results for downstream image segmentation.
>
>    - **Methodology and Protocol:** As MIMIC-CXR lacks pixel-level annotations, we adopt a surrogate task of Clavicle segmentation. We generated pseudo-ground truth masks for the training set using a specialized segmentation model (Seibold et al., 2023). For the baseline, we trained a standard U-Net on 3000 real images. We further augmented this baseline with 3000 synthetic images (totalling 6000 training samples) generated by each of the 11 models in our benchmark. To ensure rigorous comparability, evaluation was conducted on the same held-out test set (5,034 samples) used for our classification benchmarks.
>    - **Key Result:** Our result **(Table 15)** reveals an interesting contrast between classification and segmentation utility. For classification, even low-quality generators led to improved downstream performance. For segmentation, most models failed to improve or worsened downstream performance, except for the highest-quality Saana model, which improved by 2%. We attribute this to downstream classifiers’ reliance on macroscopic features (e.g., presence of a pathology), which the generative models synthesise well; in contrast to segmentation’s reliance on detailed texture, which the generative models synthesise less well. This subtle domain shift in texture makes it harder for synthetic data to be a net positive (considering the benefits of augmentation, but the drawbacks of the distribution shift). Thus, the lower-quality models tend to be unhelpful for improving downstream segmentation, and only the highest-quality Sana model is helpful.
>
> **[5] Clarification on Hyperparameters and Training Settings**
> The hyperparameters were individually tuned for each model, following the recommended settings for fine-tuning. The specific hyperparameters, such as the learning rates, are detailed in **Appendix F.1, Table 11**. Although model-specific hyperparameters like learning rates were used, all other aspects of the training protocol were kept identical to ensure a "level evaluation framework." All models were trained for exactly 20 epochs using an identical training split of 237,388 samples.
> Furthermore, we provide results from an ablation study on specific hyperparameters. In Section I, we present the results for an ablation study on the LoRA rank for large models (>1B parameters), demonstrating how performance changes with increasing LoRA ranks.
>
> **[6] Regular Updates on a Public Leaderboard**
> Yes, a public leaderboard is a planned and integral part of this work. We have updated the README file for the anonymous repository to display the public leaderboard, which can be found here: https://anonymous.4open.science/r/CheXGenBench-52F0/README.md
> Upon acceptance, we will make this leaderboard publicly available on the project webpage and maintain it with regular updates as new models are evaluated.

---

> > ### Author Response · Authors · 2025-11-27
> > **Author Response (3/3)**
> >
> > **[7] Qualitative Examples**
> > We thank the reviewer for this constructive suggestion. We have incorporated qualitative examples of synthetic generations into the revised manuscript. In **Appendix L**, we conduct an extensive analysis of the semantic properties of synthetic data and systematically examine how they differ from authentic X-ray images. **Figures 5 and 6** in the updated manuscript illustrate that synthetic X-rays maintain accurate macroscopic anatomical topology (correct placement of the heart, lungs, and other organs) while failing to replicate microscopic textural details, including bone granularity, vascular branching patterns, and intrinsic imaging noise. In contrast to the granular appearance of authentic X-rays, synthetic images exhibit enhanced contrast with an unnatural "waxy" appearance of bone structures.
> >
> > **[8] Quality Control for SynthCheX-75K**
> > We applied a stringent, automated filtration process to ensure the diagnostic plausibility of every sample included in SynthCheX-75K. This quality control procedure is detailed in **Appendix G of our paper**.
> >
> >    - **Expert VLM as a Proxy:** We utilized HealthGPT  (Lin et al., 2025), a state-of-the-art medical Vision-Language Model (VLM) with advanced reasoning capabilities, to serve as our expert evaluator.
> >    - **Expert Radiologist Prompting:** We provided HealthGPT with a meta-prompt instructing it to act as an expert radiologist. Its specific task was to evaluate the quality of a medical image based on its correspondence to the provided clinical description **(see Appendix Section G)**.
> >    - **Filtration Criteria:** The VLM classified each synthetic image into one of four categories: _**"High Quality," "Medium Quality," "Low Quality,"**_ or _**"Not Relevant"**_.
> > "High Quality" was explicitly defined as having "excellent image clarity, accurate anatomical representation, clear visibility of described pathological findings. _"Low Quality"_ or _"Not Relevant"_ images were defined as having poor image resolution, unclear anatomical structures, or content that does not correspond to the provided description. For curating SynthCheX-75K, we removed all samples labelled "Low Quality" or "Not Relevant" from the dataset, resulting in 75,649 high-quality, synthetic samples.

---

### Official Review · Reviewer_NNTd · 2025-11-01

**Soundness:** 4
**Presentation:** 4
**Contribution:** 3
**Rating:** 8
**Confidence:** 4

**Summary:**

I reviewed a paper that presents CheXGenBench, a comprehensive benchmark designed to evaluate synthetic chest X-ray generation models across three critical dimensions: fidelity, privacy, and utility. The authors aim to establish a standardized evaluation framework for medical text-to-image (T2I) models, addressing the lack of unified assessment protocols in this rapidly growing area.

The benchmark covers 11 state-of-the-art generative models and evaluates them on a large-scale clinical dataset derived from MIMIC-CXR.

In addition, the authors introduce SynthCheX-75K, a dataset of 75,000 synthetic radiographs generated with their best-performing model (Sana 0.6B), along with detailed analysis of strengths and weaknesses across different pathologies. The study reveals that current medical T2I models produce high-fidelity but clinically shallow imagesthey often fail on rare conditions, exhibit potential privacy leakage, and offer limited downstream benefits compared to real data.

**Strengths:**

The authors identify a genuine gap in how we evaluate synthetic medical image generation and address it through a comprehensive, unified benchmark that integrates three critical evaluation criteria. This multidimensional perspective is highly valuable, as most prior studies have focused solely on visual fidelity or clinical realism, while neglecting privacy and downstream impact.

The benchmark is methodologically rigorous and transparently designed. I appreciate evaluation  across a  range of metrics (FID, KID, PRDC, classification AUROC, and privacy measures). The scope and consistency of the experimental setup make the results credible and reproducible. The authors also provide a new synthetic dataset (SynthCheX-75K) and code, reinforcing the benchmark’s long-term utility for the community.

**Weaknesses:**

While the paper is very strong in scope and execution, its conceptual novelty is limited, as it primarily focuses on benchmarking and integration rather than proposing new modeling techniques or theoretical insights. The work’s contribution is therefore more infrastructural than algorithmic, which may place it slightly below the novelty threshold for top-tier venues like ICLR.

A second limitation is that the benchmark is domain-specific, focusing exclusively on chest X-rays (MIMIC-CXR). This narrow scope raises questions about generalization. for instance, whether the same evaluation framework and findings would hold for other imaging modalities (CT, MRI, ultrasound) or non-medical domains. Extending the framework to more diverse data would substantially increase its impact.

Although the privacy evaluation is interesting and well-motivated, the privacy attack scenarios remain somewhat shallow, focusing on direct pixel and latent re-identification. Stronger tests such as membership inference, model inversion, or attribute inference attacks could offer a more realistic picture of generative privacy risks.

**Questions:**

While the paper is very strong in scope and execution, its conceptual novelty is limited, as it primarily focuses on benchmarking and integration rather than proposing new modeling techniques or theoretical insights. The work’s contribution is therefore more infrastructural than algorithmic, which may place it slightly below the novelty threshold for top-tier venues like ICLR.

A second limitation is that the benchmark is domain-specific, focusing exclusively on chest X-rays (MIMIC-CXR). This narrow scope raises questions about generalization. for instance, whether the same evaluation framework and findings would hold for other imaging modalities (CT, MRI, ultrasound) or non-medical domains. Extending the framework to more diverse data would substantially increase its impact.

Although the privacy evaluation is interesting and well-motivated, the privacy attack scenarios remain somewhat shallow, focusing on direct pixel and latent re-identification. Stronger tests such as membership inference, model inversion, or attribute inference attacks could offer a more realistic picture of generative privacy risks.

---

> ### Author Response · Authors · 2025-11-27
> **Author Response (1/2)**
>
> **[1] Limited Novelty for ICLR**
> We thank the reviewer for their feedback. We respectfully disagree with the characterization of our work's novelty as limited. While we do not claim to propose a new generative architecture, our contribution is a novel evaluation framework that produces several **critical, non-obvious conceptual and algorithmic insights** that were previously obscured by inadequate benchmarking.
>
> We contribute in the following ways:
>    - **Addressing the Benchmarking Crisis in Medical Image Analysis:** Existing studies in the literature have constantly lacked a unified evaluation. We established the first benchmark for a unified assessment of fidelity, privacy, and utility. This specifically adds value through: **(i)** The novel insight that simple fine-tuning general purpose foundation models outperform existing purpose designed medical models (Sana > LLM-CXR, RadEdit). This fact was previously missed due to fragmented prior evaluations. **(ii)** Future attempts at medical generative models will for the first time be straightforwardly able to quantitatively compare against all prior work thanks to our common benchmark platform. Introducing the first reliable progress metric will accelerate future research progress.
>    - **Improving the Evaluation Metrics:** Previous work usually relied upon general purpose metrics (e.g., Inception/DenseNet FID) without class-conditional analysis. We introduced a much more thorough evaluation suite using RadDino for FID and mode-coverage (PRDC). Importantly, for the first time, we also introduce  pathology-conditional analysis. Our RadDino pathology-conditional analysis (Tab 2) shows a different set of results and conclusions to the standard unconditional FID analysis used in the literature. _This adds specific value by highlighting that the “success” of prior generative models is largely illusory and of limited clinical relevance - due to mainly succeeding to model the more frequent non-pathology class well, while falling down on many of the crucial pathology classes of interest. This identifies an important flaw in existing methods’ capabilities and an important route for future work._
>    - **Alleviating the high computational costs:** The computational barrier for rigorously training and benchmarking large-scale T2I models is substantial. We have already incurred this high cost, which for the T2I model training phase alone (excluding all evaluation) totalled approximately **800 Nvidia H200 GPU hours (~3200 A100 GPU hours)**. By open-sourcing our fine-tuned models, we add value by freely providing a foundational, ready-to-use infrastructure for the entire community.
>
> Finally, we would respectfully like to highlight that **ICLR has a strong and established precedent for publishing high-impact dataset and benchmark papers [1,2,3,4]**. This history demonstrates that the venue values foundational contributions which are essential for standardizing evaluation, ensuring reproducible progress, and identifying the critical gaps and new research directions. We believe CheXGenBench, by providing the first unified framework to rigorously assess fidelity, privacy, and utility in the biomedical domain, provides a meaningful contribution to the community in the form of a reproducible benchmark, a synthetic dataset, and new SoTA generative models.
>
> **References**
> 1. Xie, Tinghao, et al. "Sorry-bench: Systematically evaluating large language model safety refusal." International Conference on Learning Representations (2024).
> 2. Zhang, Hanlei, et al. "Mintrec2. 0: A large-scale benchmark dataset for multimodal intent recognition and out-of-scope detection in conversations." International Conference on Learning Representations  (2024).
> 3. Chen, Dongping, et al. "Gui-world: A video benchmark and dataset for multimodal gui-oriented understanding." International Conference on Learning Representations (2024).
> 4. Chang, Yapei, et al. "Booookscore: A systematic exploration of book-length summarization in the era of llms." International Conference on Learning Representations (2023).

---

> > ### Author Response · Authors · 2025-11-27
> > **Author Response (2/2)**
> >
> > **[2] Benchmark is domain-specific, generalisation to other modalities**
> >
> > **Justification for Focusing on MIMIC-CXR:** In Section 2.1, we state that the MIMIC-CXR dataset has become the _**"de-facto standard"**_ for text-to-image radiograph generation. The field is highly active, with numerous studies adopting this dataset (Bluethgen et al., 2024; Weber et al., 2023; Pérez-García et al., 2024; Lee et al., 2024; Dutt et al., 2024). However, this research has been fragmented, with "inconsistent evaluation protocols" and a lack of comparative baselines. Our primary goal was to first bring order to this specific, active domain, allowing for the rigorous, fair comparisons that uncovered our key findings (e.g., long-tail failures, privacy risks).
> >
> > **Designed for Extensibility:** While the current implementation is domain-specific, the framework is highly modular and designed for generalizability. This is a core design principle of CheXGenBench.
> >    - **Plug-and-Play Integration:** As stated in the Abstract and Section 2, the benchmark features **"seamless plug-and-play integration"** for new models.
> >    - **Decoupled Pipelines:** More importantly, Section 2 highlights that CheXGenBench features **"decoupled training and evaluation pipelines"**. Researchers can use their own training frameworks and then "automatically assess their models on over 20 standardized metrics by simply providing the generated images and a metadata file". This design means the benchmark is not hardcoded to chest X-rays. A researcher can easily adapt it to a new modality (CT, MRI, etc.).
> >
> > The core principles of assessing **Fidelity** (with domain-specific encoders), **Privacy** (re-identification), and **Utility** (downstream tasks) are universal. As we state in our conclusion, we "envision CheXGenBench to grow with new generative models and paradigms", and it serves as a robust template for generative model research evaluation in other medical (and non-medical) domains.
> >
> > **[3] Insufficient Privacy Evaluation**
> > We thank the reviewer for their suggestion regarding stronger privacy attacks. However, we respectfully point out that our evaluation _**does not**_ rely solely on shallow pixel or latent distances. Our primary privacy metric is a **Deep Learning-based Re-Identification (Re-ID) Score**, which functions as a robust, domain-specific inference attack.
> >
> > The Re-ID Score is a Strong, Learned Adversarial Attack We utilize a specialized Siamese neural network (with a ResNet-50 backbone) specifically trained to act as an attacker that predicts whether two chest radiographs originate from the same patient. This is not a simple distance metric; it is a learned model ($f_{\theta}^{re-id}$) that extracts biometric features unique to an individual’s anatomy. By calculating the probability $s_{reid}$ that a synthetic image and a real training image belong to the same patient, we are effectively performing a **strong attribute/identity inference attack**. This directly measures the risk of a model leaking specific patient identities, which is the primary concern under data protection regulations like HIPAA and GDPR.
> >
> > **Unification of Established Protocols:** A core objective of CheXGenBench is to unify the fragmented landscape of medical image generation research. Consequently, we deliberately adopted this specific Re-ID metric because it is the established standard in existing literature for quantifying privacy leakage in medical imaging (Fernandez et al., 2023; Dutt et al., 2025; Dutt, 2025). Adopting disparate or novel attack scenarios would have hindered direct comparability with prior work.
> >
> > **Complementary Role of Pixel and Latent Metrics:** We agree that reliance on a single metric can be misleading. As stated in the paper, we explicitly included Pixel Distance $l_{pix}$ and Latent Distance $l_{lat}$ as "complementary evaluation methods" precisely because "any single DL-based metric can be unreliable"6. This multi-faceted approach ensures we capture different modes of leakage, from exact memorization (Pixel Distance) to semantic identity leakage (Re-ID Score), providing a rigorous and realistic picture of privacy risks.

---

### Official Review · Reviewer_UyBi · 2025-11-01

**Soundness:** 3
**Presentation:** 3
**Contribution:** 3
**Rating:** 4
**Confidence:** 4

**Summary:**

This paper proposes CheXGenBench, the first unified benchmark for synthetic chest radiograph generation that evaluates three critical dimensions: generation fidelity, privacy risk, and clinical utility through 20+ metrics. Unlike prior work that fragmented these evaluations, this benchmark provides standardized protocols to train and evaluate 11 Text-to-Image models, identifying Sana (0.6B) as the top performer. Key findings reveal that (1) even SOTA models struggle with long-tailed medical data distributions, (2) high privacy risks exist regardless of fidelity quality, and (3) synthetic data is useful for unimodal tasks (classification) but limited for multimodal tasks (report generation). The authors also release SynthCheX-75K, a synthetic dataset of 75K samples.

**Strengths:**

**1. Comprehensive and Novel Benchmark Framework**
- First unified benchmark evaluating fidelity, privacy, and utility simultaneously for medical image generation
- Pathology-specific conditional analysis is a crucial contribution. Previous work only reported global averages, missing clinically important performance variations across different conditions
- Comprehensive evaluation of recent models (Sana, Pixart Sigma, Flux) for medical radiograph generation

**2. Rigorous Methodology and Important Findings**
- Standardized training protocol (identical data, epochs, batch size) ensures fair comparison across 11 models
- Uses domain-specific state-of-the-art encoders (RadDino, BioViL-T) instead of outdated DenseNet-121, improving evaluation reliability
- Introduces PRDC metrics to measure mode coverage.
- Multi-level privacy assessment (pixel, latent space, patient re-identification)
- Quantitatively demonstrates severity of long-tail distribution problem (correlation coefficient 0.947 between pathology prevalence and model performance). This is an important finding that reveals current models primarily memorize data distribution rather than learning to generate rare pathologies

**Weaknesses:**

**1. Unsubstantiated Claims About SynthCheX-75K and Lack of Augmentation Experiments**
- Conclusion claims SynthCheX-75K can be used "to augment existing datasets, particularly for rare conditions," but provides no experimental evidence. Table 4 only shows standalone synthetic-only training results, not real+synthetic augmentation. Given that Sana (which generated SynthCheX-75K) shows poor performance on rare pathologies like Pleural Other (FID 199.45 vs 44.60 for No Finding), how can this dataset augment rare conditions effectively? The paper's own finding of 0.947 correlation suggests all models struggle with rare pathologies, making this augmentation claim highly questionable without supporting experiments.
- Similarly claims SynthCheX-75K serves as "high-quality standalone training resource," but Table 5 shows all synthetic data underperforms real data for report generation. This contradicts the "standalone" characterization. It only works for classification, not for more complex multimodal tasks.

**2. Multimodal Task Failure Without Deep Analysis**
- Table 5 shows ALL synthetic data significantly underperforms real data for report generation (e.g., Original BLEU-1: 38.16 vs best Sana: 31.11, ~18% degradation). This is critical given "downstream utility" is a core evaluation dimension. However, the paper only provides superficial speculation ("potentially... can alleviate") without investigating root causes. Is it insufficient image-text alignment? Sana and Pixart Sigma have highest alignment scores but don't improve RRG performance. Is there a fundamental limitation in synthetic image semantics that prevents effective multimodal learning? This deserves deeper analysis.
- No experiments on real+synthetic mixture, which would be the actual practical use case for augmentation and could reveal whether synthetic data has any value when combined with real data.

**3. Incomplete Technical Details**
- Critical reproducibility details missing: LoRA alpha values not specified (Table 11 only shows learning rates). Section 3.1 mentions LoRA on attention layers might be insufficient for large models, but no experiments applying LoRA to additional layers (e.g., MLP) are conducted.
- Downstream task evaluation limited to classification and RRG. Other important tasks like object detection and few-shot learning not evaluated.

**Questions:**

**1. Evidence for Rare Condition Augmentation Claim**
- What evidence supports the claim that SynthCheX-75K can augment rare conditions? Table 2 shows Sana performs worst on rare pathologies (e.g., Pleural Other FID: 199.45). Did you conduct experiments showing that adding SynthCheX-75K to real data improves performance on rare pathologies? If not, on what basis is this claim made?
- Can you provide distribution statistics of SynthCheX-75K? Does it faithfully reproduce the long-tail distribution of MIMIC-CXR, or does it have better coverage of rare pathologies?

**2. Root Cause of Multimodal Failure**
- Why does ALL synthetic data underperform real data for RRG despite some models showing high image-text alignment? Can you provide deeper analysis beyond the current speculation?
- Did you experiment with real+synthetic mixtures for training? At what mixing ratios might synthetic data provide value?
- Could the semantic information in synthetic images be fundamentally different from real images in ways that affect caption generation but not classification?

**3. Technical Details**
- What are the LoRA alpha values? Why was LoRA only applied to attention layers, did you test applying it to MLP layers for large models?
- For SynthCheX-75K filtering (Appendix G): What percentage was classified as "Low Quality" or "Not Relevant"? How does this vary by pathology?
- Are the classification improvements (Table 4) statistically significant and generalizable to other datasets (CheXpert, PadChest)?

---

> ### Author Response · Authors · 2025-11-27
> **Author Response (1/3)**
>
> **[1] Evidence for Rare Condition Augmentation Claim**
> We thank the reviewer for their critical assessment. We agree that our initial claims regarding augmentation were not fully substantiated by the original synthetic-only experiments. To address this, we have **conducted extensive new experiments on synthetic data augmentation (Appendix K) and provided detailed distribution statistics (Appendix H)**.
>
>    - **Evidence for Augmentation Utility (Classification & Rare Pathologies)**
> We have added a dedicated section, Appendix K.1 (Image Classification), detailing the experimental evidence that supports the utility of SynthCheX-75K, and synthetic data from other T2I models, for augmenting rare conditions.
>       - **Protocol:** We constructed composite training sets by augmenting a baseline of 20,000 real radiographs from MIMIC-CXR with an equal volume (20,000 samples) of synthetic images generated by each model. The synthetic samples were conditioned on the original radiology reports of the real subset to ensure exact semantic alignment between the real and synthetic pairs. Results were reported on the same test set to maintain consistency.
>       - **Empirical Results:** Contrary to the expectation set by the poor FID scores, our results **(Table 18)** demonstrate that synthetic augmentation yields performance gains on rare classes. Models like **Sana, RadEdit, and Pixart-Sigma** outperformed the real-data baseline on minority pathologies such as **"Fracture" and "Pleural Other"**, despite low FID scores on the same pathologies. Furthermore, we also observe that low-fidelity models (SD V1-4, V1-5, etc) also provided augmentation gains, leading to an average AUC improvement of ~3% over the real data baseline. Overall, we observe that augmenting the training set with synthetic data (even with low fidelity) contributes positively to classification performance on both majority, and more importantly, minority pathologies.
>    - **Justification of Results:** We provide two explanations for these results.
>       - **The Texture-Topology Gap:** Examining the data in detail, we observe a distinct Texture-Topology Gap. Specifically, we find that T2I models generate synthetic X-rays with sufficient correctness in macroscopic topology (correct placement of heart, lungs, and other organs). However, models often fail to generate the microscopic texture, leading to poor FID scores on specific pathologies. During downstream training, the classifier (ResNet50) learns to disregard this inconsistent texture while simultaneously utilizing the topology for better generalization, leading to higher AUC scores. We confirm this visually in Figure 5 (Appendix K.1) and further expand on this phenomenon in Section Appendix L. The texture gap might be reduced in future by also fine-tuning the autoencoder used to encode the latent space (currently frozen in all experiments).
>       - **Utility Plateau:** We also observe a utility plateau, i.e. once a generative model surpasses a certain degree of semantic correctness, further improvements in texture yield limited returns for discriminative tasks like classification. This explains why models with poor fidelity (SD V1-4, V1-5) provide similar gains as models with higher fidelity (Pixart, Sana).

---

> ### Author Response · Authors · 2025-11-27
> **Author Response (2/3)**
>
> **[2] Experiments on Synthetic Data Augmentation**
> We thank the reviewer for this insightful critique. We agree that our initial "synthetic-only" evaluation did not fully capture the practical utility of these models. To address this, we have conducted a **deep root-cause analysis** and **additional augmentation experiments (Appendix K.2)**.
>   - **Rationale for Synthetic-Only Experiments:** Foundation models such as LLaVA-Rad are typically pre-trained on massive corpora (e.g., 697,000 image-text pairs) comprising both public and private datasets. Replicating this extensive pre-training regime presents significant barriers regarding data accessibility and computational resources for the broader research community. Consequently, our initial objective was to investigate whether _continual fine-tuning_ of a pre-trained checkpoint using exclusively synthetic data could serve as a resource-efficient strategy to enhance performance. We hypothesized that this setting would be the most practical setting for researchers with limited computation budgets, and without access to the private datasets used in LLaVA-Rad.
>    - **Augmentation Experiments for RRG:**  We addressed the concern regarding "Real + Synthetic" mixtures by an augmentation experiment comparing training from scratch using Real vs Real+Synthetic mixture (augmentation). Given the rebuttal turnaround time and the incomplete public availability of LLaVA-Rad training data, we focused on using synthetic data to improve LLaVA-Rad retrained on a smaller subset of data.
> Specifically, we trained LLaVA-Rad from scratch, performing both Stage 1 feature alignment and Stage 2 low-rank finetuning, using a baseline of 50,000 real samples (maintaining the same pathology ratios as in MIMIC), and augmented with 20,000 synthetic samples from each T2I model. The experimental details are provided in **Appendix K.2**, and the results are presented in **Table 19**.
>    - **Observations and Results**
>       -  **Performance Gain in RRG is Fidelity-dependent:** From **Table 19**, we find that synthetic data from T2I models with high fidelity (low FID), such as RadEdit, Pixart, and Sana, leads to consistent performance gains across all metrics over the real data baseline. However, T2I models with poor fidelity scores (SD V1-4, SD V1-5) lead to a performance degradation across all metrics. We attribute this to the fact that the inconsistency in texture of poor-fidelity T2I models possibly causes the RRG model to hallucinate or provide generic descriptions, leading to degraded scores.
>       - **SynthCheX-75K leads to consistent gains:** We also utilized the SynthCheX-75K dataset for augmentation and observed that it lead to the highest gains, confirming its utility and that high-quality synthetic data offers substantial value in RRG.
>
> **[3] Additional Medical Image Analysis Tasks (Image Segmentation)**
> We thank the reviewer for suggesting the inclusion of additional downstream tasks. We agree that a comprehensive benchmark should evaluate diverse capabilities, and we address these concerns below:
>
>    - **Justification on Initial Constraints:** As noted initially in Section 2.4 of our manuscript, our choice of tasks was constrained by the MIMIC-CXR dataset since it lacks pixel-level or bounding box ground-truth annotations. Consequently, standard Object Detection or Segmentation tasks could not be included natively.
>    - **New Addition: Image Segmentation (Appendix J):** To address the reviewer's concern regarding localization and pixel-level utility, we have added a rigorous Image Segmentation task in the revised manuscript (Appendix J). We developed the following methodology to overcome the lack of ground truth labels:
>       - **Surrogate Tasks & Pseudo-Labelling:** We established a surrogate task of Clavicle segmentation. We utilized a specialized anatomical segmentation model (Seibold et al., 2023) capable of delineating 159 anatomical regions in an X-ray. This process generated "pseudo ground-truth masks" for our training data.
>       - **Model & Protocol:** We trained a standard U-Net architecture. Our experimental protocol compared a **Real-Data Baseline (trained on 3,000 real images)** against an **Augmented Baseline (trained on 3,000 real + 3,000 synthetic images)**.
>       - **Key Results:** Our results (Table 15) highlight a sharp contrast in utility: while classification improved even with low-quality generators, most T2I models failed to improve or worsened downstream performance. Only high-fidelity models succeeded, with Sana achieving a 2% gain and Pixart-Sigma matching the baseline. We attribute this to a "Texture-Topology Gap": models effectively capture the macroscopic topology required for classification but often lack the microscopic texture fidelity needed for segmentation. Consequently, the textural domain shift in lower-quality models outweighs the benefits of augmentation, **making only the highest-quality models beneficial for segmentation tasks**.

---

> > ### Author Response · Authors · 2025-11-27
> > **Author Response (3/3)**
> >
> > **[3] Exploring the Root Cause of Multimodal Failure**
> > To recap the results so far, downstream classifier training from scratch benefits from both purely synthetic data **(Table 4)** and mixed data **(Table 18)**. For downstream RRG, continual fine-tuning degraded performance **(Table 5)**, but mixed training from scratch improved performance **(Table 19)**. Meanwhile, downstream segmentation training with mixed data provides some opportunity for performance improvement **(Table 17)**. But both RRG and segmentation improvements depended much more strongly on the choice generator compared to classification: Synthetic data from poor FID models were unhelpful, and good FID models such as Sana were required for beneficial data augmentation.
> >
> > The main “failed” result in need of explanation is therefore the continual fine-tuning (CFT) on purely synthetic data. As discussed in” Texture-Topology Gap” above, there is some residual difference between real and synthetic data in the form of texture details. We therefore attribute the failure of the CFT + Synthetic Data condition to the model forgetting knowledge from the real data (which is not revisited during our original CFT protocol) – such as the detailed textures. This forgetting adversely affects the real data testing performance. In contrast, during the mixed training of RRG, we are able to benefit from the augmented synthetic data without forgetting the real data. Similarly, the mixed training for segmentation benefits if a strong FID generative model is used.
> >
> > We swap the pure synthetic results (classification and RRG) in the main paper with the more realistic and useful mixed training results in the appendix if accepted.
> >
> > **[4] Incomplete Technical Details**
> > We thank the reviewer for requesting these technical clarifications. We have updated the manuscript to include these details explicitly.
> >    - **LoRA alpha values:** For all our experiments, we set the scaling factor alpha ($\alpha$) to equal the rank ($r=32$). We have added these details in **Appendix F.1**
> >    - **Application to MLP Layers (New Ablation):** We primarily applied LoRA to attention layers (Query, Key, Value) in the main benchmark to adhere to standard parameter-efficient fine-tuning practices. However, to address the question of whether broader application would help, we conducted a new ablation study in **Appendix I (Table 16)** evaluating LoRA placement on SD v3.5.
> >       - **Results:** We found that extending LoRA to MLP layers yielded a modest improvement in fidelity (lowering FID from 91.30 to 85.26). However, applying it to Positional Embeddings caused significant degradation (FID 99.19) 2. While Attention+MLP offers gains, the performance still lags significantly behind fully fine-tuned smaller models like Sana (FID $\approx$ 54), reinforcing our finding that model efficiency and full adaptation outweigh scaling with PEFT.
> >    - **Distribution of Image Quality in SynthCheX-75K:** We have now added the distribution of image quality per-pathology in **Table 14 (Appendix H.1)**. Overall, SynthCheX-75K constitutes 42,155 (55.72%) "Medium Quality" and 33494 (44.28%) "High Quality" samples, respectively.

---

### Official Review · Reviewer_QCXy · 2025-11-01

**Soundness:** 3
**Presentation:** 3
**Contribution:** 3
**Rating:** 6
**Confidence:** 4

**Summary:**

The paper presents a unified evaluation framework for synthetic chest radiograph generation that incorporates three main criteria: generative fidelity, privacy risk, and downstream utility. It includes around 20 quantitative metrics for these tasks, validated on 11 different text-to-image models.
The main contributions are:
- It provides a unified platform to evaluate text-to-image generative models across three major criteria.
- It presents a thorough study and extensive experiments demonstrating the applicability of different metrics for domain-specific outcomes, especially considering the long-tailed medical data distribution.
- It commits to releasing fine-tuned models and a synthetic dataset that could be valuable for the research community.

**Strengths:**

- Provides a detailed and well-reasoned study highlighting the clinical applicability of the three evaluation criteria.
-  Evaluates 11 models, including both domain-specific and general-purpose text-to-image models.
- The paper is clearly written and well-structured.
- Addresses critical domain-specific challenges, such as the long-tailed distribution in medical data, and compares commonly used evaluation backbones like DenseNet (for FID) with RadDINO.
- Commits to releasing a large-scale (20k) synthetic dataset that could significantly benefit the research community.

**Weaknesses:**

- Some parts require further explanation:
a) In Section 2.2, it is mentioned that for FID they focus on radiology-specific and biomedical domain-oriented assessment. The same section also lists other criteria such as precision, recall, density, and coverage. However, it is unclear how these metrics are computed. Are domain-pretrained classifiers or existing approaches used to obtain these measures for generated medical samples?
b) The paper emphasizes the importance of metrics that address the long-tailed distribution in medical data, but it does not explain how RadDino effectively handles such cases. If rare conditions are underrepresented in its training, would not RadDino also face similar limitations?
c) In Table 1, the “Alignment score” is reported but not described. Its definition, computation method, and relevance to the evaluation framework should be clarified.
- The distribution of samples across different categories in the 20k synthetic dataset should be discussed to better understand diversity and balance.
- In the downstream task of radiology report generation, most metrics used are general NLG-based metrics. In the medical domain, these are often insufficient for evaluating the accuracy of clinical terminology and findings. If the framework aims to serve as a unified standard, domain-specific metrics should be included. Beyond RadGraph-F1, metrics like GREEN score and RaTEScore could provide a more clinically meaningful evaluation.

**Questions:**

- In Section 2.2, it should be explained how precision, recall, density, and coverage are computed for the evaluation. It is unclear whether these metrics rely on a domain-pretrained classifier or any other existing approach tailored for medical image assessment.
- The handling of long-tailed medical data distributions by RadDino should be clarified. It is important to explain whether RadDino is trained or fine-tuned on rare disease cases and how it effectively addresses imbalance, given that such cases are inherently underrepresented.
- The definition and computation method of the “Alignment score” in Table 1 should be described to clarify its role and interpretation within the evaluation framework.
- The distribution of samples across different categories in the 20k synthetic dataset should be presented to illustrate dataset diversity and ensure transparency regarding potential biases.
- In the downstream task of radiology report generation, the choice of primarily NLG-based metrics should be justified. Since such metrics are often insufficient for clinical evaluation, the inclusion of domain-specific measures such as GREEN score and RaTEScore, in addition to RadGraph-F1, should be considered to strengthen the medical relevance of the evaluation.

---

> ### Author Response · Authors · 2025-11-27
> **Author Response (1/2)**
>
> **Clarifications on Evaluation (FID, PRDC, RadDino)**
> Thank you for highlighting the need for greater clarity regarding the computation of our mode coverage metrics. We confirm that we do not use generic pre-trained networks (like Inception V3) for these calculations; instead, we utilize domain-specific biomedical encoders to ensure the metrics capture clinically relevant features.
>
> 1. **Domain-Specific Feature Extraction (RadDino):** Just as we replaced the standard Inception V3 backbone with RadDino (Pérez-García et al., 2025) for calculating FID to avoid domain mismatch, we utilize the same high-quality embeddings from RadDino to compute the mode coverage metrics (Precision, Recall, Density Coverage). RadDino is a foundational model, trained on 838K chest X-rays, with State-of-the-Art (SoTA) performance on radiology classification and report generation tasks. This ensures that the features being analyzed for coverage and density are medically semantic rather than based on natural image statistics.
>
> 2. **PRDC Metric Computation**: Please note that Precision/Recall are not used in the sense of classifier success/error rate. FID/Precision/Recall/Density/Coverage are all assessing the unconditional quality of the synthetic data samples compared to the real data.
>    - **FID:** The simplest FID score reports the scalar distance between the real and synthetic data distributions as measured by KLD between Gaussian models of the embedding space.
>    - **Precision/ Recall:** Provide a more nuanced picture by distinguishing imperfect overlap in the KL sense as attributable to lack of quality vs lack of diversity. If the real distribution encompasses the synthetic distribution (high-precision, low recall), then samples are realistic but not diverse enough. If the synthetic distribution encompasses the real distribution (low-precision, high recall) samples are too diverse and not realistic enough.
>    - **Density/Coverage:** Aim to measure the same phenomena as precision/recall, but in a more sophisticated outlier-robust way.
>    - The basic PRDC metrics above are unconditional measures of real-synthetic data distribution similarity. We further perform conditional analysis to see how well images reflect specific pathologies by (i) “Alignment” (Tab 1), which checks the match between prompts/pathology descriptions; and (ii) Conditional-PRDC (Tab 2) which analyses the pathology-conditional PRDC metrics.
>    - All implementations follow Naeem, (ICML’20)
>
> By combining the RadDino backbone (for domain-aligned embeddings) with the PRDC framework (for manifold assessment), and conditional analysis (for image-pathology correspondence), we provide a rigorous evaluation of how well the synthetic samples cover the complex, long-tailed distribution of medical pathologies.
>
> **How does RadDino handle rare pathologies?** We would like to highlight an important clarification in this aspect that establishes RadDino as a highly capable feature extractor. RadDino is trained on a **significantly larger and more diverse superset of data**. While the generative models in CheXGenBench were trained exclusively on the MIMIC-CXR dataset (approximately 237k samples) to simulate a standard medical T2I training setup, RadDino was pre-trained on a much larger corpus of **838k chest radiographs**. Crucially, as RadDino’s training corpus is approximately 3.5 times larger than the dataset employed in CheXGenBench, pathologies that are rare in our training distribution are significantly better represented within the evaluator's diverse, multi-source pre-training data. Consequently, RadDino acts as a robust image descriptor that allows reliable measurement of the extent to which images of any pathology are similar between real and synthetic data. We have reworded Section 2.2 (“Improving metric Reliability in CheXGenBench”) to better explain feature extraction using RadDino.
>
> More generally, we remark that standard practice prior to our work was to use either OOD feature extractors (ImageNet pre-trained), or outdated ID feature extractors trained on much smaller datasets (Medical DenseNet-121, ~100k images) for synthetic data quality assessment. So prior work suffers much more severely from the reviewers’ concern of questionable reliability of image-similarity measurement, especially in the long-tail data case. Our evaluation advances he metric reliability substantially by adopting a modern, robust and in-domain feature embedding trained on larger-scale data for the first time.

---

> > ### Author Response · Authors · 2025-11-27
> > **Author Response (2/2)**
> >
> > **[1] Clarification on the Alignment Score (Table 1)**
> > We clarify that this metric is the global cosine similarity computed between an encoded image and text caption using BioViL-T (Bannur et al., 2023), a domain-specific vision-language model.
> > Our Alignment Score is essentially a domain-specific CLIPScore metric [1], a standard, reference-free evaluation metric used for all “CLIP-style” models. BioViL-T is architecturally based on CLIP [2], utilizing modality-specific encoders to project image (X-Ray) and text (clinical caption) embeddings into a shared semantic space. The score is computed as the global cosine similarity between these embeddings, quantifying how well the generated image aligns with the clinical text caption without requiring a reference image.
> >
> >    - **Relevance to Evaluation:** This alignment score is an integral part of our evaluation, as it directly measures the extent to which the synthetic radiographs are faithful to the prompt, which is a necessary condition for clinical utility. A low score signifies that the generative model has failed to adhere to the conditioning text, omitting or hallucinating the specific medical details required by the prompt. High alignment is thus a prerequisite for the synthetic data to be considered reliable for any downstream clinical or research application. **We have added a more descriptive explanation of the Alignment Score in Section 2.2 (“Improving metric Reliability in CheXGenBench”) of the updated manuscript.**
> >
> > **[2] Additional Metrics for RRG Evaluation (GREENScore and RaTEScore)**
> > We thank the reviewer for this suggestion and agree that adding these new metrics would make the evaluation more comprehensive. **We have incorporated these metrics in the benchmark and augmented Table 5 (RRG results).**
> >
> > **[3] Pathology Distribution in Synthetic Datasets**
> > Thank you for this suggestion. Adding the distribution of pathologies in SynthCheX-75K would enhance the utility and understanding of the diversity of the dataset. **We have added these details in Appendix Section H.1 (Table 14)** in the updated manuscript. The pathology distribution in SynthCheX-75K follows that of the original MIMIC-CXR dataset, which would enable researchers to adopt it as an external test set to validate the robustness of new models.
> >
> > **References**
> > 1. Hessel, Jack, et al. "Clipscore: A reference-free evaluation metric for image captioning." Proceedings of the 2021 conference on empirical methods in natural language processing. 2021.
> > 2. Radford, Alec, et al. "Learning transferable visual models from natural language supervision." International conference on machine learning. PmLR, 2021.

---

### Author Response · Authors · 2025-11-27
**Shared Response (1/2)**

We thank the reviewers for their constructive feedback and high assessment of our work’s scope and execution. We are encouraged that the reviewers recognize CheXGenBench as a **"comprehensive and novel benchmark"** (R2) that addresses a **"genuine gap"** (R3) and provides **"high methodological quality"** (R4). The reviewers’ insightful questions regarding the utility of synthetic augmentation and the root causes of multimodal failure have driven us to conduct significant additional experiments. We have updated the manuscript with **three new Appendices (J, K, L)**. Below, we address the common themes. For ease of readability, the updates in the manuscript are **highlighted in red.**

**Synthetic Data Augmentation Experiments:** We have performed additional experiments for studying the effect of synthetic data augmentation for (1) Image classification and (2) Radiology Report Generation (RRG) in **Appendix K**. For Image classification, we observe that synthetic augmentation consistently matched or outperformed the real-data baseline. Crucially, we observed performance gains on minority pathologies such as "Fracture" and "Pleural Other" using models like Sana and RadEdit **(Table 18)**. For RRG, performance gains were observed only in the case of high-fidelity models (RadEdit, Pixart-, Sana), while augmenting with synthetic data from low-fidelity models (SD V1-4, V1-5) had a detrimental effect. Our results showcase that RRG is a texture-sensitive task and benefits from improved upstream performance.

**Additional Medical Image Analysis Tasks:** We have expanded our benchmark to Image Segmentation, focusing on the illustrative task of segmenting Clavicles from X-rays (Appendix J). To overcome the lack of ground truth in MIMIC-CXR, we generated pseudo-labels for our training data using a specialized anatomical segmentation model. The results show that augmenting downstream model training with data from the highest-fidelity model (Sana) can improve performance.
Analysing our results, we make an observation that we term the "Texture-Topology Gap": Synthetic images often correctly reflect their prompt on the macro scale, but differ from real images in subtle textural details, which is reflected in fidelity scores. In downstream tasks where subtle textures are less critical (classification), it is easier to improve performance by augmenting with data from various generative models. In contrast, downstream tasks where such subtle details are more important (RRG and segmentation), are more sensitive to the quality of the upstream generative model in terms of whether they benefit from synthetic data augmentation.

**Incomplete Technical Details:** We have updated the manuscript with the following clarifications about several technical details.
1. **Alignment Score, RadDino, and PRDC (Section 2.2):** We clarified that the Alignment Score is a domain-specific CLIPScore, calculated as the global cosine similarity between image and text embeddings within the BioViL-T latent space. Additionally, PRDC metrics are computed using high-quality RadDino embeddings to ensure clinical relevance, rather than generic ImageNet features.
2. **LoRA Technical Details (Appendix I):** We specified that Alpha ($\alpha$) was set equal to Rank ($r$) throughout our experiments. We also perform new ablations extending LoRA to MLP layers and find that it only leads to marginal gains.
3. **Training Hyperparameters:** We have clarified in the updated manuscript that while learning rates were tuned per architecture for convergence, the training budget (20 epochs) and data split were identical across all models to ensure a fair comparison.
4. **Details on SynthCheX-75K:** We have expanded the manuscript to include comprehensive dataset statistics. **Appendix H.1 (Table 14)** details the per-pathology frequency, confirming faithful reproduction of the long-tailed clinical distribution. Additionally, **Appendix G** now reports the overall and per-pathology distribution of image quality ratings derived from the filtration process, ensuring full transparency regarding the rejection rates of synthetic samples.

---

> ### Author Response · Authors · 2025-11-27
> **Shared Response (2/2)**
>
> **A Deeper Analysis into Multimodal Task Failure:** We conducted a deeper analysis investigating the performance degradation originally reported for the RRG task. Through quantitative experiments and qualitative observations, we posit a Texture-Topology Gap that characterizes current synthetic radiographs. While generative models successfully reproduce Macroscopic Topology (anatomical coherence sufficient for classification), they struggle with Microscopic Texture, often exhibiting a "waxy" appearance lacking the high-frequency noise of real data. This textural domain shift confuses the visual encoder in multimodal models when using a fixed projector (continual fine-tuning), leading to feature misalignment and performance degradation. However, we demonstrate that this can be resolved by training from scratch, which allows the model to learn a joint embedding for both real and synthetic textures, effectively unlocking the utility of synthetic augmentation.
>
> **Incorporation of Clinically-Aware Metrics for RRG:** We have expanded our RRG evaluation suite to include two domain-specific metrics: GREENScore and RaTEScore. We have updated Table 5 (Main Paper) and Table 19 (Appendix K) to report these metrics.

---

### Meta-Review · Area_Chair_Uyo3 · 2026-01-06

**Summary:**

This paper presents a unified benchmark for fidelity, privacy and utility of synthetic chest radiography. Overall, it receives diverse opinions with various level of confidences. The reviewers have precisely pointed out some of the issues in this paper. For example, the solution seems to be general but the authors only work on Chest Radiographs. I agree that  benchmarking of data synthesis is important. But this is indeed a general task. It is not very well motivated to handle it for chest data specifically. On the other side, the motivation to unify the fidelity, privacy and utility is not clear.  The reviewers have also raised some concerns on the experimental results. Overall, I am not supportive to this work. I would suggest the authors to submit the work to medical domains where the paper can be better evaluated.

**Reviewer Concerns:**

Some of the concerns remains, for example, the work shall be evaluated in general tasks. While some reviewers recommended high scores, they also raise some important points. For example, Reviewer NNTd pointed out that this  work primarily focuses on benchmarking and integration rather than proposing new modeling techniques or theoretical insights. Therefore, he/she placed it slightly below the novelty threshold for top-tier venues like ICLR. This reviewer also raised concern on the narrow scope and also whether the same evaluation framework and findings would hold for other imaging modalities (CT, MRI, ultrasound) or non-medical domains. Reviewer nMkj echoed with similar concerns.

**Reviewer Scores:**

The reviewers have been given some high scores, despite that they have pointed out some major issues. Therefore, I do not think whether they will further improve the scores would be a main factor here.

---

### Decision · Program_Chairs · 2026-01-26

Reject